# Blaming luck, claiming skill: Self-attribution bias in error assignment

**Naoyuki Okamoto**[1,2], **Michael Taylor**[3], **Takatomi Kubo**[1,4], **Shin Ishii**[2,5,6], **Benedetto De Martino**[1,3☯]*, **Aurelio Cortese**[1,7,8☯]*

**1** Computational Neuroscience Laboratories, ATR Institute International, Kyoto, Japan, **2** Graduate School of Informatics, Kyoto University, Kyoto, Japan, **3** Institute of Cognitive Neuroscience, University College London, London, United Kingdom, **4** Graduate School of Science and Technology, NAIST, Nara, Japan, **5** Neural Information Analysis Laboratories, ATR Institute International, Kyoto, Japan, **6** International Research Center for Neurointelligence (WPI-IRCN), Institutes for Advanced Study, University of Tokyo, Tokyo, Japan, **7** Department of Biomedical Engineering, Sungkyunkwan University, Suwon, South Korea, **8** Center for Neuroscience Imaging Research, Institute for Basic Science, Suwon, South Korea

☯ These authors contributed equally to this work.
* benedettodemartino@gmail.com (BDM); cortese.aurelio@gmail.com (AC)

## Abstract

Mistakes are valuable learning opportunities, yet in uncertain environments, whether a lack of reward is due to poor performance or bad luck can be hard to tell. To investigate how humans address this issue, we developed a visuomotor task where rewards depended on either skill or chance. Participants consistently displayed a self-attribution bias, crediting successes to their own ability while blaming failures on randomness, an effect that influenced their subsequent decisions. Computational modelling revealed two underlying mechanisms—a distorted perception of ability and a positivity bias in the skill condition. Notably, while distorted self-perception shaped behaviour, it did not affect confidence; instead, self-attribution bias led to overconfidence in external blame. These findings suggest a more complex picture in which self-attribution biases arise from both perceptual distortions and post-decision evaluations, highlighting the need for an interplay between experimental design and computational modelling to understand behavioural biases.

## Author summary

When we fail, it's often unclear whether the cause is a lack of skill or simply bad luck. In everyday life, where many factors interact, it can be difficult to determine whether we are responsible for specific outcomes. To investigate how people handle this uncertainty, we designed a visuomotor task in which feedback sometimes reflected actual performance and sometimes was random. On each trial, participants had to infer whether the feedback was meaningful or due to chance. Participants consistently exhibited a self-attribution bias: they credited positive

**Data availability statement:** Behaviour data and code used for the analyses are available at https://github.com/ATR-decnef/WAM_publish; https://github.com/BDMLab.

**Funding:** Authors received funding from the following sources. Ikegaya Brain-AI Hybrid ERATO grant (JPMJER1801) from the Japan Science and Technology Agency: A.C JSPS KAKENHI grant-in-aid grant number JP22H05156: A.C. JSPS KAKENHI grant-in-aid grant number JP22H04998 and JP22H05163: S.I. Innovative Science and Technology Initiative for Security Grant Number JPJ004596, ATLA, Japan: A.C. the Japan Trust International Research Cooperation Program of the National Institute of Information and Communication (NICT): B.DM., A.C. JST SPRING Grant Number JPMJSP2110: N.O. Funders played no role in the study design, data collection and analysis, decision to publish, or preparation of the manuscript.

**Competing interests:** The authors have declared that no competing interests exist.

outcomes to their own ability, while attributing negative outcomes to randomness. Computational modelling revealed an asymmetry in how they updated their beliefs—positive feedback influenced their inferences more strongly than negative feedback. This "positivity bias" led participants to emphasise successes and discount failures in the skill condition. Interestingly, confidence was not driven by distorted self-perception but was higher when the feedback was random, suggesting overconfidence when blaming external factors. By combining behavioural experiments with computational modelling, our study shows that self-attribution bias can arise also from perceptual distortions, in addition to biased post-decision evaluation. These findings provide insight into the mechanisms behind humans' biased interpretations of success and failure.

## Introduction

Using feedback (both positive and negative) to update beliefs and adjust behaviour lies at the heart of reinforcement learning [1,2]. This powerful learning strategy enables animals, including humans, to determine the correct course of action directly from their environment through trial and error. Algorithms based on RL have proven essential in driving the ongoing AI revolution (e.g., AlphaGo and AlphaFold) [3,4]. However, this learning strategy relies on the tacit assumption that humans (or artificial agents) can easily identify the source of an error leading to the lack of a reward. While this holds in most laboratory settings, where scenarios are designed with clear objectives and straightforward causes of errors, the complexity of the real world introduces multiple, often hidden, sources of errors. To learn efficiently, humans (and advanced AI systems) must discern whether a negative outcome stems from a genuine error or merely bad luck, such as randomness in the environment [5,6].

Here, we developed a visuomotor task to test how humans respond to errors caused by their performance or by factors independent of it. We found that, although participants can generally determine the source of an error, they tend to attribute the cause of a mistake to the environment while crediting themselves for success. This behavioural bias appears to manifest a general behavioural tendency well documented in psychology, known as self-attribution bias, often also called self-serving bias [7].

For example, in finance, self-attribution bias occurs when a trader attributes gains to their own skills but blames negative results on bad luck or other external factors [8,9], leading to overconfidence and excessive risk-taking as traders overestimate their abilities [10]. Similarly, in educational settings, students often credit high grades to their own intelligence and effort while teachers attribute these successes to their own effective teaching methods, whereas students blame poor grades on unfair tests while teachers blame inadequate students' preparation [11,12]. Notably, self-attribution bias has been found consistently across different population strata, albeit with substantial variations due to age, cultural background and psychopathology [13]. In the context of mental health, depression has been linked with reduced

self-attribution bias and the development of learned helplessness [14–16]. A recent study proposed self-attribution bias as a computational mechanism that might protect against learned helplessness, counterbalancing higher learning rates from negative feedback [17].

However, most work on this phenomenon is limited to psychological traits [18] and their motivational underpinnings [19]. Only recently have a number of studies started to address how the valence of outcomes shapes learning in the context of reinforcement learning and Bayesian inference. While some authors have found that people harbour higher learning rates for positive outcomes [20–22], others have shown the opposite effects [23,24]. To reconcile these findings, some have suggested that the direction of valence-dependent learning asymmetries is due to beliefs about the causal structure of the environment [25].

In this study, we examined two competing mechanisms for self-attribution bias. One possibility is that the bias arises from post-hoc outcomes evaluation. Individuals tend to give greater weight to positive feedback (successes) and downplay negative feedback (errors) in conditions in which they have control over the environment (i.e., in the skill condition). Alternatively, the bias may stem from an inflated internal representation of one's abilities, which distorts how individuals perceive the environment.

To resolve this question, we incorporated several new key features into our experimental setup. First, inspired by two recent studies [17,26], we diverged from most previous work by employing a sensorimotor task rather than the more commonly used multi-armed bandit tasks, in which factors beyond mere choice play no role. This created a scenario in which the participant's actual motor performance had a measurable effect on the feedback, with performance being continuously monitored prior to feedback. Second, we utilised a computational model incorporating a perceptual representation of one's motor skills as an explicit threshold parameter. This model enabled us to test the effects of the alternative cognitive mechanisms described above. Finally, by measuring confidence on a trial-by-trial basis, we were able to examine the complex relationship between metacognition and the underlying cognitive strategy of credit and blame assignment.

## Results

### Task, error assignment and discovery of a novel self-attribution bias

Our new visuomotor task was inspired by the arcade game 'whack-the-mole' and modified to create scenarios in which rewards were sometimes tied to participants' motor performance ('skill' condition) and sometimes independent ('random' condition). In each trial, participants had to quickly hit (within 800 ms) a cartoon mole that popped out of one of several ground holes shown on a touchscreen tablet. After hitting a mole, participants received a positive or negative feedback, which depended on the mole hit location in the 'skill' condition and was random in the 'random' condition. They then reported whether they believed this feedback was due to their action or mere randomness (see Fig 1A, inference choice on the hidden task state 'skill' or 'random'). Finally, participants reported their confidence about the accuracy of their decision (Fig 1B). The task state ('skill' or 'random') was hidden and changed at random time points, unbeknownst to participants (see Fig A in S1 Text) for sequence durations between switch points). Thus, participants had to weigh multiple sources of uncertainty to correctly infer the ongoing task state: uncertainty about the hidden rule and uncertainty about their motor movements in hitting the mole. Overall, participants' (N = 66) inference accuracy was above the theoretical chance at the group level (mean accuracy 0.65, ± 0.10, Wilcoxon signed rank test against chance level 0.5: Z = 6.63, $P < 0.001$). Participants with at-chance performance or with fixed confidence ratings were excluded from further analyses (see Methods for full exclusion criteria). Thus, N = 51 participants were included in all subsequent analyses. While the task resulted in varying degrees of inference accuracy at the individual level, participants well adapted to the changing hidden state of the task (see two example participants in Fig 1D). The task was calibrated such that participants would obtain positive feedback with similar probabilities in both 'random' and 'skill' task states. This was achieved by setting the threshold for the positive feedback in the skill state individually as the median distance of a participant's' hit location distribution

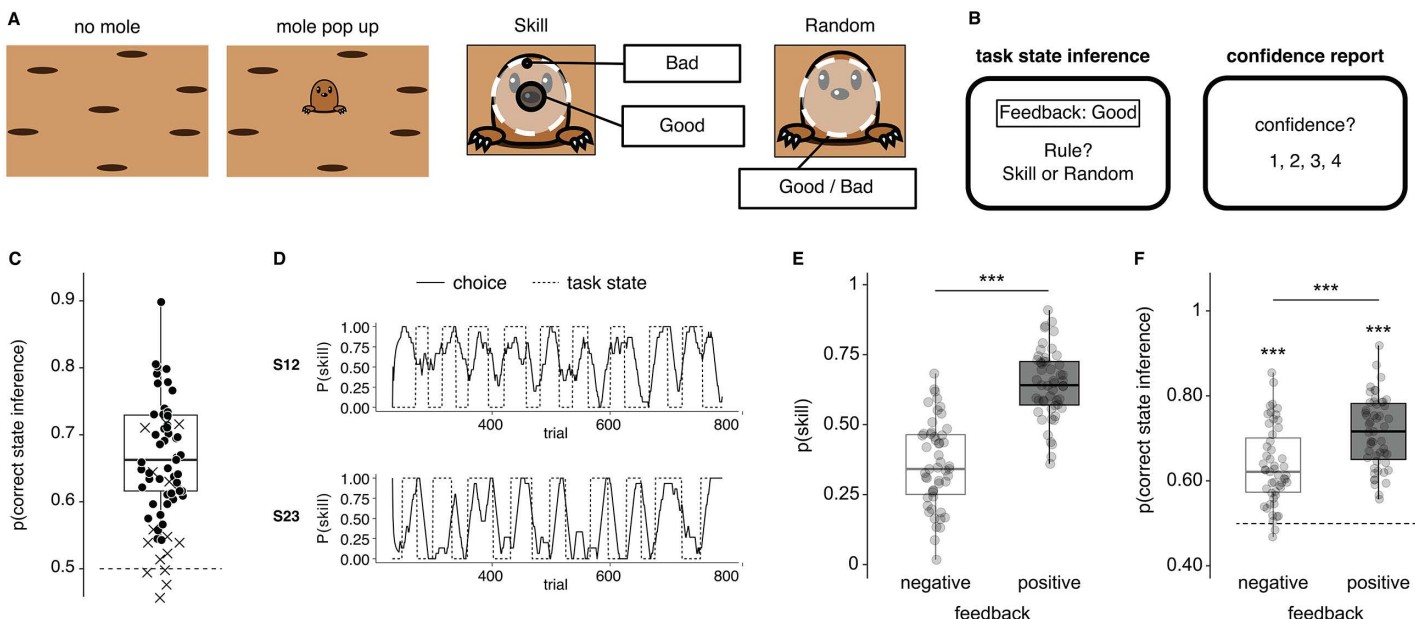

**Fig 1. Task design, general behaviour results and self-attribution bias. (A)** To score points in the task, participants had to hit (with a finger, on a touchscreen) a mole that quickly popped up and down. Following a hit, participants received the feedback (positive/negative). There were periods in which the feedback depended on the participant's skill to hit the centre of the mole (skill condition) and periods in which the feedback was random (random condition). The number of trials between change points followed a gamma distribution with a mean duration of 28 trials. The illustration was created with Adobe Illustrator 29.6 (Macintosh). **(B)** After seeing the trial's feedback, participants reported whether they thought they were in the skill or random condition. Next, participants rated their confidence in the correctness of their inference. **(C)** Participants' overall accuracy in inferring the task state (N = 66). Crosses represent participants excluded from further analyses (N = 15, see methods), while black dots represent included participants (N = 51). **(D)** Example time courses of task state inference from two participants. Behaviour trajectories were smoothed with a backwards time window of size n = 15 trials. **(E)** Participants' probability of choosing the task state 'skill' as a function of feedback ('negative', 'positive'). **(F)** Participants' probability of correctly inferring the hidden task state as a function of feedback ('negative', 'positive'). In panels C, E and F, dots represent individual participants, and box plots the median and interquartile ranges. N = 51 human participants, *** $P < 0.001$ (Wilcoxon signed-rank test).

during the preliminary score prediction task. Note that there was nonetheless a significant difference between the states (mean ratio of obtaining the positive feedback in the skill state was 0.56 while the one in the random state was 0.50, Fig B in S1 Text). Overall, there was little difference between the two hidden states in participants' inference rate of the actual condition (see Table A in S1 Text, Wilcoxon signed rank test on diagonal elements, i.e., true positive and true negative rates: $Z = 1.87$, $P = 0.061$).

Participants displayed a clear self-attribution bias. In short, they chose the 'skill' state significantly more often when they received positive feedback following a mole hit than when they received negative feedback ($Z = 5.99$, $P < 0.001$, Fig 1E). In turn, participants' probability of making a correct state inference also differed depending on the feedback received, with overall higher accuracy in trials whose feedback was positive compared with trials whose feedback was negative ($Z = 5.22$, $P < 0.001$, Fig 1F). Note that inference accuracy was significantly above chance in both cases (test for inference accuracy > 0.5; negative feedback: $Z = 6.14$, $P_{FDR} < 0.001$, positive feedback: $Z = 6.21$, $P_{FDR} < 0.001$). Similarly, participants had slower choice reaction times following negative vs positive feedback (Fig C1 in S1 Text).

Higher inference accuracy following positive feedback could have reflected a nonspecific task confound, making positive feedback easier to evaluate. To check for this, we first tested the effect of the feedback (negative vs positive), the true state of the task (skill vs random) and their interaction on participants' inference accuracy using a repeated measures two-way ANOVA (Fig 2A). This analysis revealed a significant interaction ($F_{(1, 50)} = 127.3$, $P < 0.001$), as well as a significant

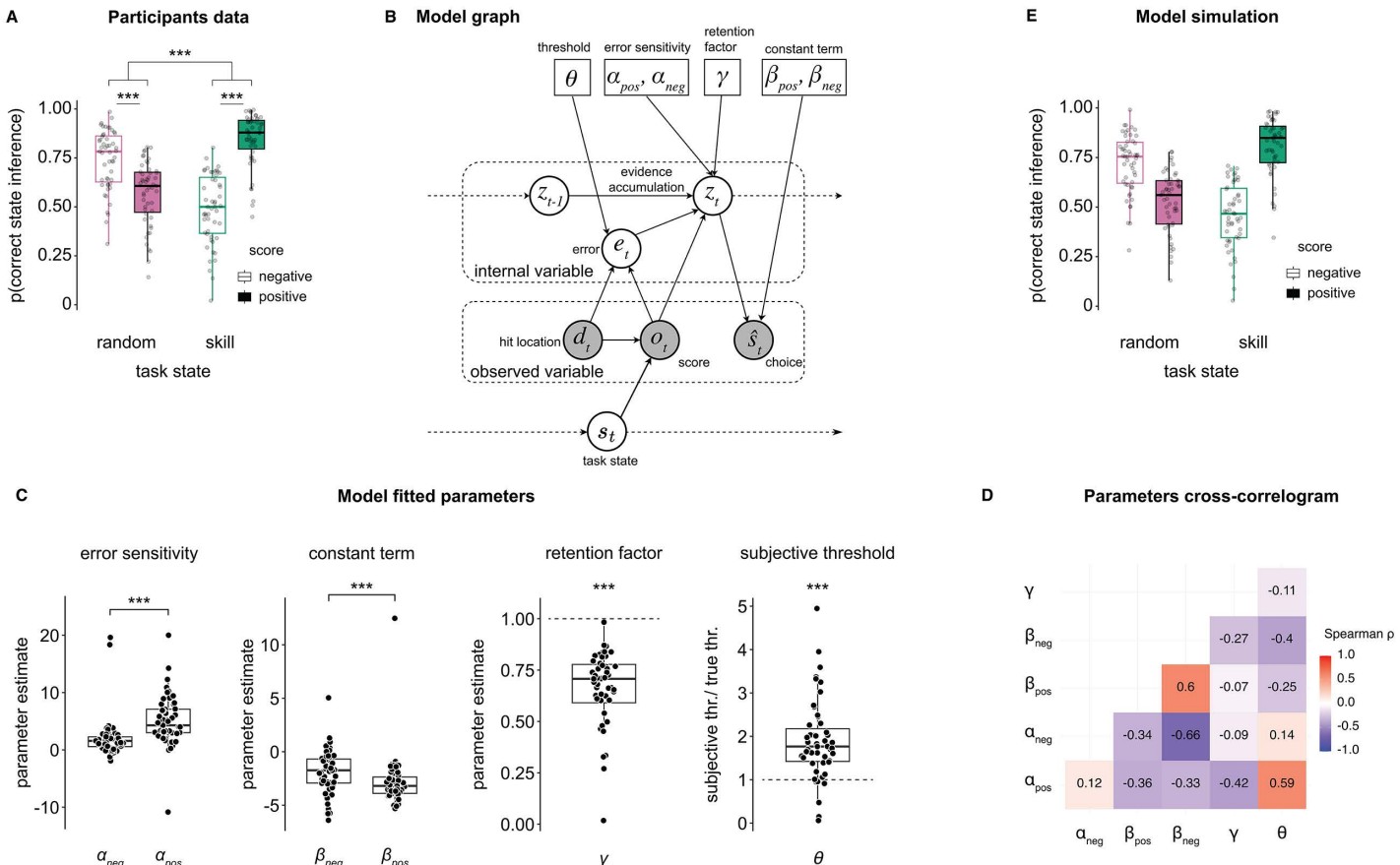

**Fig 2. Inference accuracy and computational model. (A)** Participants' probability of correctly inferring the hidden task state as a function of the feedback obtained ('negative', 'positive') and the true task state ('random', 'skill'). The effects of feedback and task state and their interaction on participants' inference accuracy were statistically evaluated with repeated measures two-way ANOVA. Dots represent individual participants, boxplots the median and first/third quartiles, and whiskers the minimum and maximum of the data range. **(B)** Graph illustration of the computational model. The model features a set of observable variables [the distance of the hit location from the mole centre $d_t$ and the feedback obtained $o_t$] and latent variables (evidence accumulation $z_t$ and the error $e_t$). The model's output is the inferred state (choice) $\hat{s}_t$. The key parameters that control the model's behaviour are the agent's subjective threshold between the centre and edge of the mole (θ), the error sensitivity modulating the effect of the error (feedback-dependent, $\alpha_{pos}$, $\alpha_{neg}$), the retention factor modulating the evidence accumulation (γ), and a constant term modulating the decision boundary (separately for positive and negative outcomes, $\beta_{pos}$, $\beta_{neg}$). **(C)** Model parameters are fitted through negative log-likelihood minimisation. The four panels show, from left to right, the feedback-dependent error sensitivity α, the feedback-dependent constant term β, the retention factor γ, and the ratio between the subjective threshold θ and the true threshold. **(D)** Parameters cross-correlogram. We used Spearman rank correlation across all parameters' distributions (each parameter was fitted at the individual participant level) to compute the cross-correlogram. **(E)** Model simulations of p(correct state inference). Simulations based on the estimated parameters and the original trial time courses. Each dot represents one participant/agent for all scatter plots, *** $P<0.001$.

main effect of feedback ($F_{(1, 50)} = 60.2$, $P<0.001$), while the effect of task state was not significant ($F_{(1, 50)} = 0.17$, $P=0.68$). Post-hoc pairwise comparisons showed drastic differences in inference ability, with higher accuracy following negative compared to positive feedback in the 'random' state ($Z=-5.01$, $P_{FDR}<0.001$) but a mirrored effect of higher accuracy following positive compared to negative feedback in the 'skill' state ($Z=6.14$, $P_{FDR}<0.001$). Accordingly, negative feedback led to higher accuracy in the 'random' state compared to the 'skill' state ($Z=-4.86$, $P_{FDR}<0.001$), while positive feedback led to higher accuracy in the 'skill' state compared to the 'random' state ($Z=5.56$, $P_{FDR}<0.001$).

We then tested whether participants' mole-hitting patterns provided additional insight into their behavioural strategy. To do this, we analysed the hit locations, precisely the distance from the mole centre, in three ways: (i) statically, (ii) over time

and (iii) in relation to previous feedback and decisions. First, we confirmed that hits were overall closer to the centre when they resulted in positive compared to negative feedback (as expected given the task design, Fig D2 in S1 Text). Interestingly, we found that where participants hit the moles in relation to the centre differed depending on the objective task state (see Fig D1-D2 in S1 Text), hits were closer to the centre in the skill state than the random state). Furthermore, on a more granular level, feedback (negative, positive) and task state inference (skill, random) impacted where participants hit the mole on the following trial (Fig D3 in S1 Text), main effect of feedback $F_{(1, 50)} = 159.68$, $P < 0.001$; main effect of inference choice $F_{(1, 50)} = 24.96$, $P < 0.001$; interaction between inference choice and feedback: $F_{(1, 50)} = 71.85$, $P < 0.001$). Participants hit closer after negative feedback than after positive feedback, whichever rule they chose (random: $Z = -5.32$, $P_{FDR} < 0.001$; skill: $Z = -6.21$, $P_{FDR} < 0.001$). Together, these results suggest that participants dynamically adjusted where they hit. However, there was no evidence that they did so strategically, in which case we should have seen an effect only in the 'skill' state and not in both, as reported here.

### Behavioural signatures of self-attribution bias and computational modelling

How does self-attribution bias lead to incorrect error attribution, and what are its computational underpinnings? More specifically, at what stage does this error attribution occur? To address these questions, we tested two alternative cognitive mechanisms.

The first mechanism operates at the feedback or decision level. In this scenario, participants tend to neglect negative feedback, much like positivity or confirmation bias [20,21]. The second mechanism involves an inflated estimation of one's abilities (motor in this task); as a result, participants are less inclined to attribute errors to themselves.

To test these two possibilities, we developed a computational model of our task in which feedback derives from the joint effect of the hit location (i.e., distance from the centre of the mole) and the hidden task state (Fig 2B). The model simulates participants' beliefs about the hidden state using a leaky error-evidence accumulator updated on every trial by an error term computed from the mismatch between the expected hit location (centre, edge) and the feedback obtained (positive, negative). Within the model, the more positive the accumulated belief value is, the more likely participants will choose the random state.

After the agent strikes a mole at a distance $d_t$ from its centre, the model compares this distance to a subjective threshold $\theta$. If $d_t < \theta$, it expects positive feedback; otherwise, it expects negative feedback. A binary error signal $e_t$ is generated—0 if the actual feedback matches the expectation, and 1 if it doesn't. This error updates the belief $z_t$, weighted by a sensitivity parameter $\alpha$ (which depends on the feedback type), and modulated by an integration factor $\gamma$ to discount prior beliefs. Finally, the probability of a latent state is computed by applying a sigmoid function to the updated belief with a feedback-dependent constant term $\beta$. Differences in error sensitivity $\alpha$ or constant term $\beta$ between positive and negative feedback would suggest a feedback/decision-dependent mechanism of self-attribution bias. Instead, perception of motor ability is captured by the parameter $\theta$, modelling participants' subjective threshold, i.e., the perceived boundary coding a switch in feedback from positive to negative in the 'skill' state. A threshold $\theta$ larger than the true threshold would indicate an inflated perception of motor ability. Note that while the model assumes that participants have limited access to the precise hit location, this assumption is restricted to a binary distinction—whether the hit landed within the centre area or outside it. The model does not otherwise rely on precise spatial information (i.e., the exact distance from the centre), which participants could not access. However, it does incorporate the idea that participants have an internal representation of the centre area—delimited by the hidden threshold—and are therefore able to form a general sense of whether a hit was relatively close to or far from the centre.

We first fit the model's free parameters to participants' behaviour data through a negative log-likelihood minimisation procedure. Parameters' fits (Fig 2C, 2D) revealed several interesting behaviour features. First, the error sensitivity $\alpha$ was larger for positive feedback compared to negative feedback ($Z = 5.08$, $P < 0.001$), indicating that participants weighted more feedback that resulted from positive feedback (in line with work on confirmation bias in reinforcement learning

[20,27]). Similarly, the constant term β was larger (more negative) for positive feedback compared to negative feedback (Z = -4.07, P < 0.001). Note that the constant term regulates the decision boundary in reporting 'skill' and 'random', and in our model, a more negative constant term shifts the decision boundary towards 'skill' inference choice. Thus, this result highlighted the overall tendency for positive feedback to lead to skill choices. Third, the error evidence accumulation process was leaky rather than lossless, as the retention factor γ was significantly smaller than 1 (Z = -6.21, P < 0.001), reflecting the noisy nature of the participants' error evaluation process. Fourth, and most critical to test one of our hypotheses, the ratio of the subjective threshold θ and the true threshold was significantly larger than 1 (Z = 5.48, P < 0.001), indicating that participants consistently overestimated their ability to get positive feedback by liberally assessing the mole central hit zone (giving positive feedback in the 'skill' hidden state). Participants' bias in assigning positive feedback to their own ability resulted from a distortion of their perceived ability, in this case, motor ability.

Our model comparison analysis further strengthened these results. We compared the full model with alternative, simpler versions. In particular, we were interested in the direct comparison with a single feedback-independent error sensitivity or constant term, true threshold, and no retention factor. The full model better accounted for participants' choice strategies (mean AIC: 463.9, all other models ΔAIC > 19, see Fig E in S1 Text). In addition, the model comparison further highlighted that the perceptual inflation(subjective threshold θ) played a larger role than overall positivity bias (error sensitivity α and constant term β) in determining behaviour, given the larger AIC difference from the full model when this parameter was fixed to the true threshold value ("true threshold", mean AIC: 525.5) than when either error sensitivity or constant term were set to symmetric across feedback type ("common error sensitivity", mean AIC: 483.3; "common constant term", mean AIC: 502.0, Fig E in S1 Text).

Next, we simulated new choice data using the parameters obtained from the fitted full model. We verified that the model could accurately capture the key behavioural signature of self-attribution bias in error assignment (Fig 2E, see also Fig F in S1 Text). These results were extended with a new set of simulations with pre-set parameter values, such as fixing the subjective threshold θ to the true value, fixing α and β to be equal for both positive and negative feedback, revealing that under different parameter regimes, behaviour changed drastically. Under these circumstances, the model failed to replicate participants' inference accuracy patterns (Fig G in S1 Text). In particular, simply fixing θ to the true value cancelled a significant portion of the self-attribution bias (absence of bias in the 'random' state and minimised bias in the 'skill' state, Fig G2 in S1 Text). The oracle model (true threshold, single error sensitivity and single constant term) displayed high overall inference accuracy across all task states and feedback without biases, as expected (Fig G8 in S1 Text).

Finally, we validated the model fitting procedure by performing a parameter recovery analysis with simulated data. Note that although some correlations between parameters were relatively high (Fig 2D), we were able to individually recover all parameters with good reliability in the parameter recovery analysis (Fig H in S1 Text). Results thus confirmed the model's robustness, with correlations between fitted parameters and recovered parameters at $r \geq 0.74$. Refitting simulated data with the entire set of models (full, simpler alternatives) highlighted the robustness of the model. The best fit model was the same as the underlying ground-truth, generative model (e.g., we recovered the full model when the full model had been used to generate simulated data, Fig I in S1 Text). Together, these simulations, parameter and model recovery analysis demonstrate the robustness of the model, but also its ability to capture multiple contributing factors in participants choice behaviour and self-attribution bias.

## Self-attribution bias determines future behaviour strategy

Participants adapted smoothly to hidden switches in the task state (from 'random' to 'skill' or vice versa). As shown in Fig 3A, the probability of making a correct inference dropped below 0.4 immediately after a task state switch, then recovered and stabilised over the following 7–8 trials. Because we observed that inference accuracy varied with the hidden task state (i.e., Fig 2A), we first asked whether the direction of the switch ('random'→'skill' vs. 'skill'→'random') differentially influenced participants' inference accuracy. We specifically focused our statistical analysis on trials following a switch (in Fig 3A, trials 1–8). A

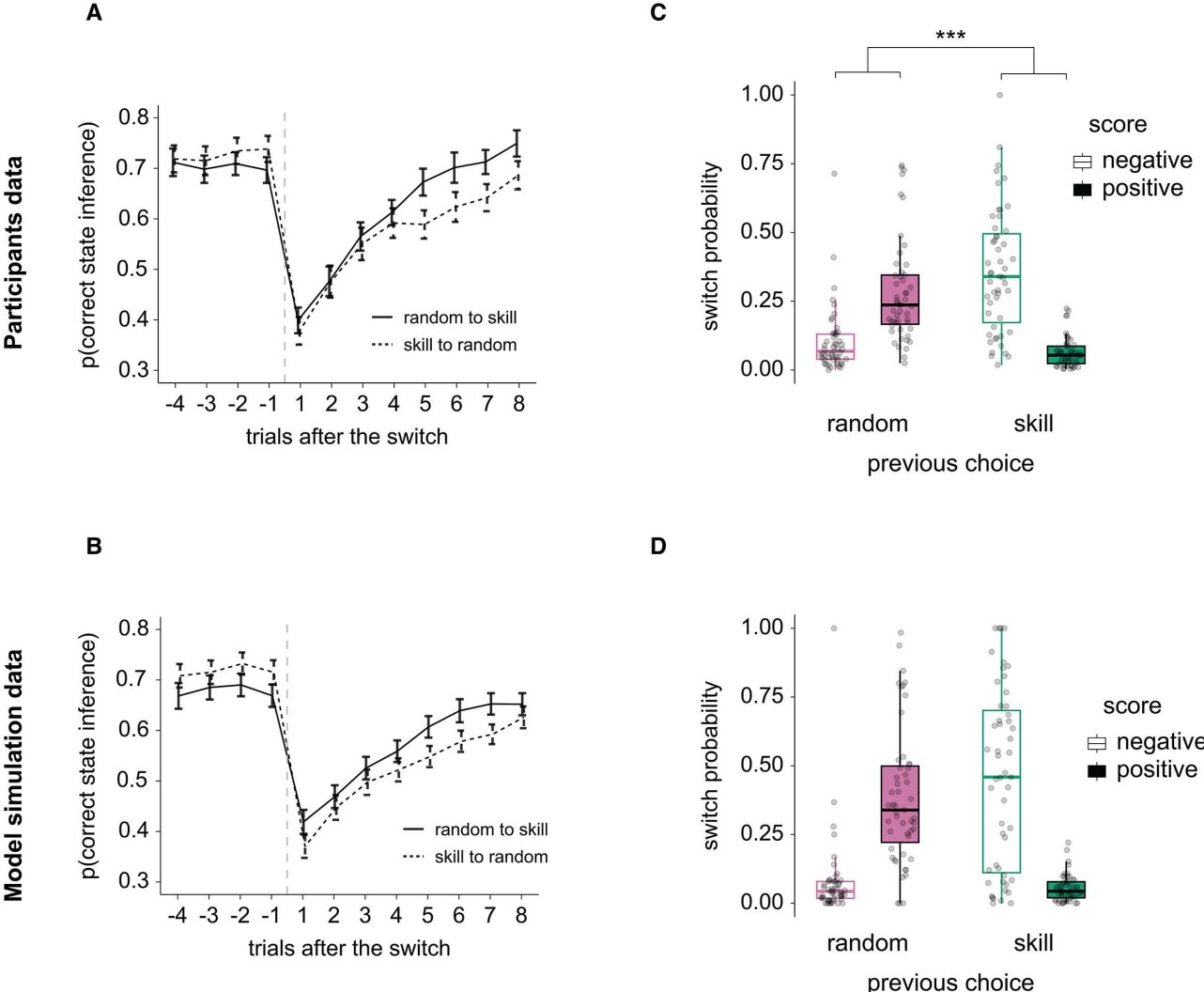

**Fig 3. Inference choices around task change points and effect on subjective switch probability. (A)** Participants' probability of correctly infer-ring the hidden task state as a function of the trial from the objective task state switch and the direction of the switch (solid line 'random → skill', dotted line 'skill → random'). The main effects of trial and switch direction and their interaction on participants' inference accuracy were statistically evaluated with repeated measures ANOVA. Solid/dotted lines represent the group average, and the error bars represent the standard error of the mean. N = 51 participants. **(B)** Same as in A, but with artificial data generated from a simulation. N = 51 simulated agents. **(C)** Participants' probability of switching their choice (e.g., from random to skill or skill to random) as a function of the feedback obtained ('negative', 'positive') and their choice in the previous trial ('random', 'skill'). The main effects of feedback and choice and their interaction on participants' switching probability were statistically evaluated with repeated measures two-way ANOVA. Dots represent individual participants, boxplots the median and first/third quartiles, and whiskers the minimum and maximum of the data range. **(D)** Same as in C, but with artificial data generated from a simulation. For all plots, N = 51 participants (A, C) or simulated agents (B, D); *** P < 0.001.

repeated-measures ANOVA revealed no significant difference in inference accuracy between the two switch directions ($F_{(1, 50)}$ = 1.70, $P = 0.20$), beyond the expected general increase in performance over trials from switch time (main effect of time: $F_{(1, 50)}$ = 66.6, $P < 0.001$) and no significant interaction ($F_{(1, 50)}$ = 1.84, $P = 0.079$). None of the pairwise comparisons between the two trajectories resulted in significant differences (all $P_{FDR} > 0.1$). Notably, model simulations closely captured this behavioural time course (Fig 3B). See Fig J in S1 Text for simulations with different parameter regimes.

We examined how self-attribution bias might shape participants' subsequent choices. Because the hidden task state ('random' vs. 'skill') and the feedback (positive vs. negative) jointly influenced correct inferences, we hypothesised that belief about the task state (previous choice at trial *t-1*) and the current feedback (at trial *t*) would together determine whether participants switched their choice at trial *t* (e.g., from 'skill' to 'random', or vice versa, Fig 3C). Indeed, there was a strong interaction between current feedback and previous choice on the probability of switching ($F_{(1, 50)}$ = 75.43, *P*<0.001). We also found a main effect of the feedback ($F_{(1, 50)}$ = 15.06, *P*<0.001) but no effect of the previous choice alone ($F_{(1, 50)}$ = 1.57, *P*=0.22). Specifically, if participants had reported 'random' in the previous trial, receiving negative feedback on the current trial led to a significantly lower chance of switching than receiving positive feedback ($Z$=-5.11, $P_{FDR}$<0.001). This pattern reversed if participants had previously chosen 'skill': after positive feedback, the probability of switching was lowest ($Z$=5.81, $P_{FDR}$<0.001). In other words, participants switched more often if they believed the hidden state was skill but received negative feedback or if they believed it was random and received positive feedback. These findings demonstrate how self-attribution bias—attributing successes to one's own skill and failures to external randomness—directly shapes behaviour and decision strategies. The model captured this strategy-updating pattern (Fig 3D).

## Confidence and its dissociation from inference accuracy

During the task, participants also reported confidence judgments about the correctness of their inferential choices. Using a repeated measures two-way ANOVA, we confirmed a significant main effect of choice correctness ($F_{(1,50)}$ = 176.5, *P*<0.001) and of feedback valence ($F_{(1,50)}$ = 39.69, *P*<0.001) on confidence, but no interaction ($F_{(1,50)}$ = 1.22, *P*=0.27). Decision confidence in correct trials was higher than in error trials (Fig 4A, $Z$=5.01, *P*<0.001). Participants' confidence was highest after receiving positive feedback ($Z$=5.01, *P*<0.001, Fig 4B), an effect consistent with previous findings [28,29]. Moreover, we found that confidence accurately reflected task uncertainty with respect to the subjective threshold parameter signalling a shift between positive and negative feedback in the skill state (Fig 4C). Confidence was lowest near the boundary, and highest at the extreme (centre, or edges). This result supports the validity of the θ parameter as a subjective threshold.

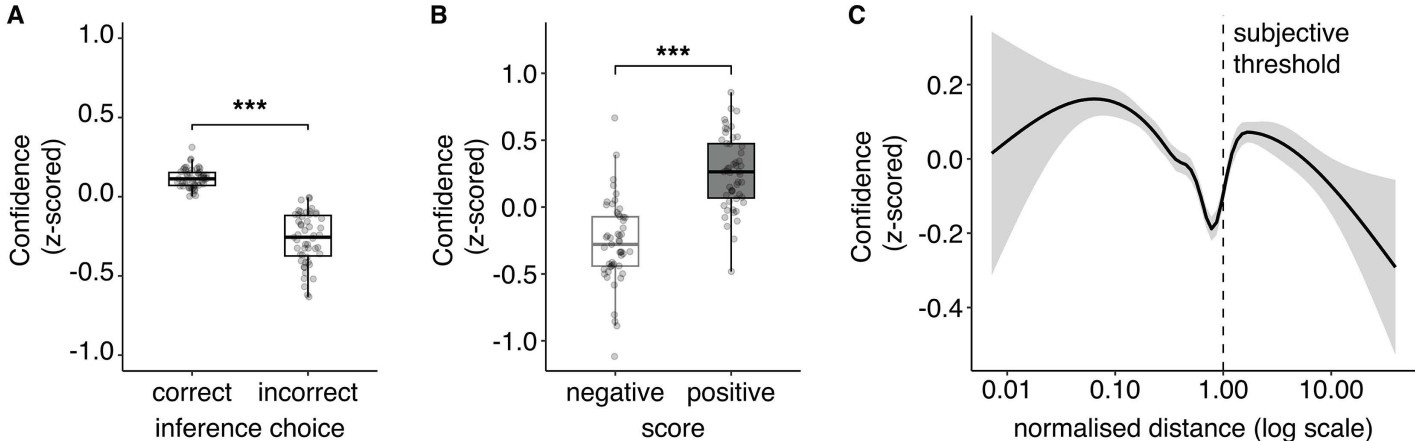

**Fig 4. Confidence about inference correctness. (A)** Participants' confidence about the task states inference. Data plotted separately for correct and error trials, highlighting the typical signature of decision confidence, with higher confidence in correct trials compared to error trials. **(B)** Confidence as a function of the feedback obtained ('negative', 'positive'). As with task state inference accuracy, confidence was higher following positive feedback. Dots represent individual participants, boxplots the median and first/third quartiles, and whiskers the minimum and maximum of the data range. **(C)** Local polynomial regression (LOESS, quadratic function with smoothing parameter=0.75) between confidence and log-transformed hit distance from the centre (normalised with the subjective threshold, indicated by the vertical dotted line). The black solid line represents the mean across participants, and the shaded area the standard error of the mean. The x-axis is on a base-10 logarithmic scale. Confidence was z-scored within participants. *** *P*<0.001.

PLOS Computational Biology

We then tested whether feedback (negative vs positive) and hidden task state ('skill' vs 'random') might differentially impact confidence judgements (repeated measures two-way ANOVA, Fig 5A). We detected a main effect of feedback ($F_{(1,50)}$ = 44.75, $P < 0.001$, i.e., participants reported higher confidence in positive than negative feedback: 'random' state: $Z = 4.11$, $P_{FDR} < 0.001$; 'skill' state: $Z = 5.39$, $P_{FDR} < 0.001$), a main effect of task state ($F_{(1,50)}$ = 16.68, $P < 0.001$), and a significant interaction ($F_{(1, 50)}$ = 10.1, $P = 0.0025$). The interaction reflected the main effect of the task state on confidence in the presence of negative feedback, in which confidence was higher in the 'random' than 'skill' state ($Z = -4.99$, $P_{FDR} < 0.001$, simple effect) but not of positive feedback, in which confidence was similar in both task states ($Z = -0.56$, $P_{FDR} = 0.58$, simple effect).

Note that we do not have access to model-based confidence judgements for the computational model since it was fit and optimised simply on the inference choice data, not confidence. However, as a straightforward proxy of confidence for the model-based simulations, we used negative entropy (entropy represents the uncertainty in the model-based decision; see Fig K in S1 Text confirming the correspondence between participants' confidence

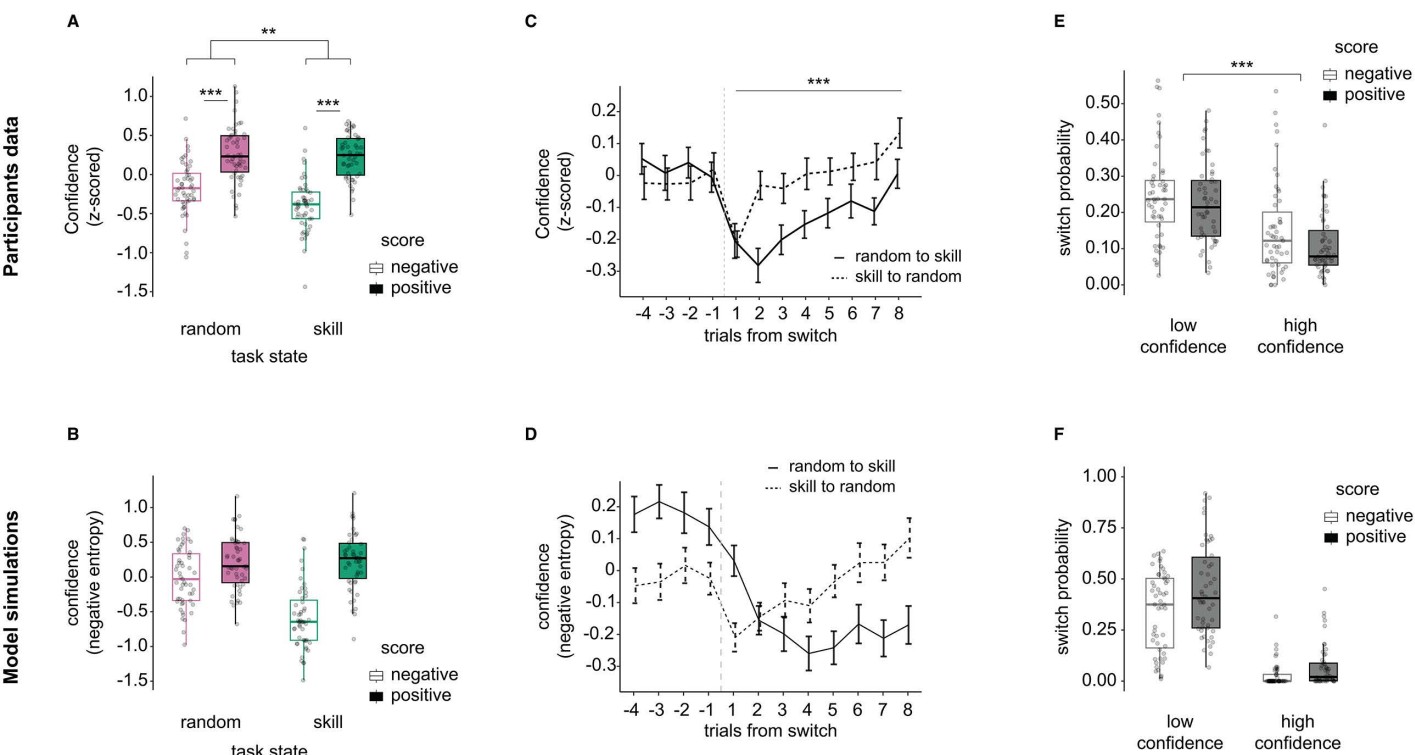

**Fig 5. Confidence about the correctness of the inference. (A)** Participants' confidence about their hidden task state inference as a function of the feedback obtained ('negative', 'positive') and the true task state ('skill', 'random'). The main effects of feedback and task state and their interaction on participants' inference accuracy were evaluated with repeated measures two-way ANOVA. For all plots, dots represent individual participants, and box plots represent the median and interquartile ranges. **(B)** Same as in A, but with artificial data generated from a simulation. Note that the model was only optimised using inference decision data, not confidence. However, since the model decision module was based on a logistic function, we used the z-scored negative entropy about the model's hidden task state inference decision as a proxy for confidence. **(C)** Time series of participants' confidence around the switch (e.g., from random to skill or skill to random). The solid line represents the mean of average confidence within participants, and the bar is the standard error of the mean. **(D)** Same as in C, but with artificial data generated from a simulation. **(E)** Switch probability of the inference choice as a function of the feedback obtained ('positive', 'negative') and the confidence ('low', 'high', binarised within participants). The main effects of feedback and confidence and their interaction on switch probability of the inference were evaluated with repeated measures two-way ANOVA. **(F)** Same as in E, but with artificial data generated from a simulation. ** $P < 0.01$, *** $P < 0.001$.

judgements and the model's negative entropy). Even though its primary purpose was not to model confidence, our computational model still closely replicated the confidence pattern found in participants (Fig 5B, see also Fig L in S1 Text). Further simulations with different parameter regimes reinforced the qualitative differences between inference accuracy and confidence uncovered earlier (Fig M in S1 Text, persistent valence effect on confidence independent of task state).

The valence effect on confidence suggests a potential metacognitive dissociation between choice accuracy and confidence (e.g., comparing Figs 2A vs. 5A) that depended on the task state-feedback condition. Overall meta-d' (a measure of how well confidence tracks decision accuracy [30]) did not differ across task states ($Z = 0.056$, $P = 0.96$) nor feedback ($Z = 0.58$, $P = 0.57$, see Fig N1-N2 in S1 Text). Participants however displayed a specific drop in metacognitive ability in trials with negative feedback in the skill state (see Fig N3-N4 in S1 Text). Self-attribution bias thus appeared to make participants overconfident not in attributing successes to themselves but in attributing blame to external causes (i.e., negative feedback blamed on environmental randomness).

On a trial-by-trial level, confidence reflected changes in task uncertainty, with participants giving their lowest ratings on average in the trials immediately following a change in the hidden task states (Fig 5C). However, confidence judgements differed starkly from the state inference accuracy trajectories around these time points (i.e., Fig 3A). The pattern of confidence judgements was asymmetric with respect to the direction of the transition (random→skill, skill→random), with a significant main effect of switch direction ($F_{(1, 50)} = 12.7$, $P < 0.001$), besides a general main effect of trials post-switch ($F_{(1, 50)} = 8.19$, $P < 0.001$), and a lack of interaction ($F_{(1, 50)} = 1.49$, $P = 0.17$). In short, reported confidence decreased when the hidden state changed, suggesting that participants generally noticed task environment changes, even without direct, explicit feedback about the task state or their choices. Nevertheless, confidence recovered faster when the hidden state switched from skill to random than when it switched from random to skill. Although exaggerated, the computational model displayed a similar difference, with a slow confidence recovery rate following a random→skill task state transition (Fig 5D). Note that plotting a longer time-scale (from -16 trials pre-switch to +15 trials post-switch) shows that both participants and model had similar tendencies: overall higher confidence in the random condition (Fig O in S1 Text). Simulations done with specific parameter settings revealed one more important piece of the puzzle (Fig P in S1 Text). All simulations in which the subjective threshold parameter θ was fixed to the true value displayed confidence patterns that were very similar to participants' own confidence ratings (Fig P2, P4, P6, P8 in S1 Text). This result suggests a dissociation between confidence and choice, in which choice is affected by the perceptual distortion of self-attribution bias, but confidence is not.

In line with previous work [31], we finally sought to verify whether confidence informs future choices. Participants had a higher probability of switching when they reported low confidence (main effect of confidence $F_{(1, 50)} = 73.70$, $P < 0.001$) but also when the feedback obtained was negative (main effect of feedback $F_{(1, 50)} = 6.25$, $P = 0.016$, no significant interaction $F_{(1, 50)} = 1.50$, $P = 0.28$, Fig 5 E). The model captured these effects. These results are reminiscent of previous findings suggesting confidence regulates behaviour updates [32,33] but also further highlight the strength of valence distortions in controlling behaviour.

## Discussion

This study aimed to understand how people attribute or misattribute negative feedback to their own perceptual errors or from environmental noise (i.e., bad luck). We compared two main hypotheses regarding the origin of the self-attribution bias we isolated in the context of a controlled visuomotor task. The first hypothesis considers that bias arises from a self-confirmatory mechanism, similar to positivity or confirmation biases [20,29,34]. In this view, individuals outweigh positive feedback (successes) and underweight negative feedback (errors) at the moment of the feedback evaluation. Instead, the alternative hypothesis states that the bias results from a systematic misestimation of one's own abilities—in this case, motor ability—leading individuals to attribute errors to external factors due to an inflated perception of personal

competence. Importantly, each explanation implies different underlying cognitive processes, necessitating distinct interventions if one aims to develop a behavioural nudge to reduce the bias.

Using behavioural and computational modelling, our study provides converging lines of evidence that a self-attribution bias can emerge from a systematic inflation of one's perceived ability. We show that the parameter θ that controls the perceptual threshold in our task is greater than the true threshold (on average, almost twice as large), indicating that participants consistently overestimated their motor skills by liberally considering the mole central hit zone (giving positive feedback in the skill hidden state). Simulations and model comparison analyses validated this result and rejected alternative interpretations. First, the model with a fixed threshold was significantly worse than the model with the subjective threshold at explaining participants' data ($P < 0.001$, average $\Delta AIC = 61.6$). Second, the model with a fixed threshold was also significantly worse than alternative models with feedback-independent error sensitivity ($P < 0.05$, average $\Delta AIC = 23.5$) or constant term ($P < 0.001$, average $\Delta AIC = 42.2$).

This pattern of results implies that self-attribution bias appears to be an intrinsic component of basic sensory processing related to the self, influencing behaviour directly from the outset of the action, beyond a post-hoc rationalisation for negative feedback. There is growing evidence for early modulation of perceptual processes by a variety of contextual constraints such as environment, task and even cognitive demands [35–37].

However, our model-based analysis has portrayed a more complex picture in which the overestimation of the skill (while key to triggering the bias) is not the only process that plays a role in generating the effect we report. We also show that, in line with work on confirmation bias in the context of reinforcement learning [20,27], participants weighted more feedback that resulted from positive feedback, in line with our first hypothesis. The confirmation (or positivity) bias might even act as a compensatory mechanism that enhances behavioural performance under the presence of self-attributional perceptual distortions (e.g., see Fig G3 in S1 Text) for simulation results with an agent harbouring perceptual inflation of ability but no confirmation/positivity biases in error sensitivity and decision parameters, showing reduced inference accuracy in the skill condition). This speculation was supported by a second preliminary piece of evidence, in that the extent of the perceptual distortion in one's ability correlated with the strength of the positivity/confirmation bias (see Fig Q in S1 Text).

A recent study revealed dynamic, reciprocal links between outcome attributions and self-beliefs, consistent with the attribution-self-representation cycle theory [17]. Participants were more likely to update their beliefs about their abilities when they attributed outcomes to themselves rather than external factors. With increasing skill estimates, participants took more credit for wins and less blame for losses. The authors also found a higher learning rate for negative than positive feedback. A different set of studies also reported greater error sensitivity following negative outcomes, driven by a self-attribution bias, as participants needed good results to earn rewards and thus learned more from failures [25,38].

In contrast, our study found greater error sensitivity (α) for positive feedback. This difference likely stems from our task structure: participants had to infer the task state (random or skill) from probabilistic, implicit feedback. Negative feedback was ambiguous—arising from both motor errors in the skill condition and accurate actions in the random condition—whereas positive feedback in the random state clearly signalled a mismatch between belief and outcome. As outcomes were only partially informative of hit location, and further obscured by motor noise, participants might have weighted more diagnostic (positive) feedback more strongly. Thus, our study highlights how task structure and outcomes interact with self-attribution bias to shape error assignment and learning, which may also underlie apparently opposing results [39,40].

Our model formalises this dynamic using two parameters per feedback type: α, modulating error accumulation, and β, biasing beliefs toward the skill state. These parameters exert opposing forces: larger α increases the accumulated belief evidence in favour of the random state, while more negative β shifts belief towards skill attribution. While our parameter fits revealed that $\alpha_{pos} > \alpha_{neg}$ (suggesting higher learning from surprising successes), this was accompanied by $\beta_{pos} < \beta_{neg}$,

indicating a concurrent attribution bias toward skill in positive feedback (since this parameter shifts the baseline of the fluctuation regardless of the existence of error). These opposing forces likely balance each other in shaping choices, and indeed, we observed a negative correlation between $\alpha$ and $\beta$ within each feedback type. This reflects a trade-off between error sensitivity and prior bias, not parameter redundancy. If both parameters encoded the same latent process (i.e., if there were information leakage), they would not be independently recoverable—a possibility ruled out by our parameter recovery analyses (Fig H in S1 Text).

Additionally, we found a positive correlation between $\theta$ (the subjective threshold) and $\alpha_{pos}$, suggesting that participants who overestimated their precision were more likely to interpret unexpected successes as random events. In this view, $\alpha_{pos}$ might act as a compensatory mechanism in which higher $\alpha_{pos}$ corrects for inflated self-beliefs. Overall, our findings expand on prior work by showing that attribution biases may emerge not only from motivational factors, but also from structural features of the task and distortions in perception.

How does self-attribution bias influence future choice? We found that misjudgements regarding perceptual abilities led participants to adjust their overall behavioural strategy. Specifically, they were more likely to change their choice if they believed the hidden state was skill-based but received negative feedback or if they believed it was random and received positive feedback. An effect that our model captured. These findings illustrate how erroneously attributing successes to personal skill and failures to external factors can significantly affect subsequent behavioural strategies, often resulting in future suboptimal decisions [22]. In a similar context, expectation about controllability has been linked to enhanced learning from failure [26]. There, high expectations of controllability increased learning from negative outcomes, helping subjects quickly recognize unachievable goals and protect their optimistic beliefs about control. Our approach may thus provide a well-controlled framework to study self-attribution bias in the broader context of marketing, managerial and financial decision-making [41,42].

Using confidence ratings collected at the trial-by-trial level, we studied the relationship between self-attribution bias and metacognitive introspection. First, we confirmed that confidence, as predicted by classical signal-detection theory (SDT), was higher for correct inference decisions than for incorrect ones [30,43]. Interestingly, we also detected an effect of valence on confidence – participants were more confident after positive feedback. While standard SDT does not predict these effects, they have been reported in the literature [28,29,44]. These results are consistent with a post-hoc positive bias, reflecting a positive post-decision evaluation of one's actions. However, we also detected a strong effect of hit location on confidence reports (Fig 4C). In short, confidence was lowest near the model-estimated subjective threshold between centre and edge, and highest at the extreme (centre, edges). This result provides additional, albeit indirect, evidence that self-attribution bias might operate also at a perceptual level.

Although we designed and fit the computational model only on inference accuracy, we found it could also capture key aspects of participants' confidence judgments. Further simulations uncovered an important yet surprising underlying phenomenon (Fig P in S1 Text). Participants' confidence was closer to the model harbouring no perceptual distortion, implying a mechanistic dissociation between choice and confidence in self-attribution bias. This dissociation resonates with previous findings with obsessive-compulsive disorder patients, where individuals form accurate confidence judgements but fail to use this confidence to update action policies [32]. From a different perspective, recent work found similar dissociations between choice accuracy and metacognitive evaluations that depended on the task context--simple perceptual decisions versus complex economic decisions [45].

After a hidden task state switch, confidence recovered faster in the skill-to-random transition (as opposed to the random-to-skill transition). This result was present even in the simulations with the oracle agent (using the true threshold, harbouring no biases in error sensitivity or constant term), suggesting that, in the context of our task, detecting an error was easier than confirming the absence of an error [46–49]. Finally, self-attribution bias affects confidence in a particular way, leading participants to be overconfident in attributing blame to external causes but not success to themselves.

Ultimately, while we interpret the subjective threshold parameter θ as reflecting participants' internal estimate of their own motor precision, we acknowledge that it can also be viewed as encoding beliefs about task difficulty—i.e., participants had precise perception of where they hit, but they had imprecise inference of the boundary between good and bad performance. Teasing apart perceived ability from inferred task demands is inherently challenging, as both likely contribute to the setting of θ. However, we suggest that the interpretation as an expression of the task difficulty appears less plausible in our design. Specifically, participants were trained during a practice session (80 trials) with explicit feedback on the true state, which should have allowed them to learn the overall difficulty and structure of the task. We therefore propose that θ may primarily capture the *remaining* uncertainty: participants' perceptual uncertainty about their own motor performance. The nature of our task, which combines inference with sensory-motor control, inherently introduces such uncertainty, preventing perfect access to the actions performed. Our data is consistent with this interpretation. We found that participants who overestimated their motor accuracy in the initial score prediction task also exhibited larger objective thresholds, indicating lower accuracy in hitting the centre, during the main task (Fig R in S1 Text). This finding underscores the positive correlation between motor ability and bias in perception of motor ability.

Note that, in Fig R in S1 Text, we also found a modest underestimation in high-performing individuals. This pattern is consistent with established psychophysical effects, in which individuals tend to be overconfident about their successes when they perform poorly [50,51]. In contrast, higher perceptual ability can lead to more conservative confidence bias placement [52], and increased signal strength can reduce overconfidence by tightening confidence thresholds [53], potentially resulting in a modest underestimation of their performance. Our findings replicated this systematic skew in perception, and it is reasonable to infer that the same perceptual skew was present during the main task and affected participants' rule inference. However, this evidence remains indirect. We cannot entirely rule out the possibility that the threshold parameter also encodes cognitive evaluation of the task difficulty to hit the centre. Further research is required to explicitly decouple these two factors.

In summary, our research shows that self-attribution bias, a common yet subtle cognitive distortion, may arise not only from processes of post-decision evaluation but also from perceptual distortions. An intriguing possibility is that perceptual distortions might be rooted in unequal or suboptimal evidence accumulation, an effect that eye-tracking and pupil-linked arousal processes could measure [54,55]. Our results offer fresh perspectives on how behavioural biases can emerge through multiple factors. They underscore the necessity of employing rigorous experimental designs alongside computational modelling to identify and understand these biases effectively. Such an approach will be crucial for developing targeted interventions to reduce or mitigate the impact of self-attribution bias.

## Methods

### Ethics statement

The Advanced Telecommunications Research Institute International (Japan) Institutional Review Board approved the study protocol (ethics number #763). All participants provided written informed consent before beginning the experiments and were instructed they could withdraw their participation at any time.

### Participants

We recruited 66 participants (23 female, mean ± SD age = 29.1 ± 8.8 years old, range 20–51 years old) to take part in this behaviour study. Participants were paid 3,000 JPY for a 1.5-hour participation. Participants were excluded from further analyses if they met any of the following exclusion criteria: (i) their accuracy about task state inference in the main task was less than 0.54—the minimum above-chance accuracy threshold given the total number of trials, determined with a binomial test; (ii) choosing the same option ('skill', or 'random') in more than 80% of the trials; (iii) reporting the same level of confidence in more than 80% in the trials. Thus, we selected 51 participants for the analyses reported in this paper (15 female, mean ± SD age = 28.2 ± 7.7 years old, range 20–46 years old).

## Task

The Whack-A-Mole arcade game inspired the main task: a player's goal is to hit a mole while it briefly pops out from one hole out of a set of N holes on a board. We chose this game as it is a game of skill in which the player has to make rapid, precise movements to hit a target, and it would lend itself well to feedback manipulations based on skill vs. luck (environment randomness).

Participants completed four task sessions: (1) mole hit and score prediction practice, (2) score prediction, (3) rule prediction practice, and (4) rule prediction main task. The first session was for participants to get used to hitting moles; the second was to evaluate the motor accuracy of participants and their ability to judge where they hit the mole and to calibrate the difficulty of the main task accordingly; the third was to let participants learn the structure of the main task, and the fourth was the main task. All tasks used a Windows PC with a touchscreen, and moles appeared and disappeared quickly, one by one, from one of the seven holes (see Fig 1). Each presentation started with a 100 ms period during which the mole moved up from the hole, followed by a 600 ms period during which the mole appeared in full, and finally, a 100 ms period during which the mole disappeared by descending in the hole. Participants were also told to use only the index finger of their right hand to touch the screen. Note that the period during which participants could hit a mole ranged from 200 ms from the start to the end (800 ms) of the mole presentation. If they missed a mole, the trial was aborted, but no penalty was applied, and another mole popped up at a different location after a delay of 200 ms.

**Mole hit and score prediction practice.** Participants were instructed to hit the mole as close to its centre as possible. When they hit a mole, concentric circles were displayed with a score of 0–100 points corresponding to the hit location. This practice session lasted 50 trials.

**Score prediction.** As in the initial practice, participants were instructed to hit the centre of the moles. After a hit, they predicted the points (0–100) with a slider without further clues (i.e., concentric circles were not shown to participants). Participants also reported how confident they were about the correctness of their prediction on a Likert scale with four levels, with one being the lowest (guess) and four being the highest (certain). This prediction session lasted 100 trials. We used the hit location distribution to calculate the individualised threshold that would determine the main task's binary feedback (positive/negative) in the skill condition. This threshold was calculated as the median hit location, i.e., the distance from the mole centre.

**Rule prediction practice.** Participants were instructed to hit the mole quickly before it disappeared. There were two task states: 'skill' and 'random'. In the 'skill' state, the hit location directly determined the feedback. If participants hit closer to the centre than the threshold, they obtained positive feedback, while they obtained negative feedback if they hit further away but within the mole area. The threshold was individually calibrated for each participant from the preliminary score prediction task, such that ~50% of mole hits would fall within the mole's central area (hit location < threshold). Importantly, participants did not know the exact location of the threshold. In the 'random' state, feedback were entirely random, i.e., there was an equal probability of getting a positive or negative feedback at any hit location. Participants had to infer the current task state based on the displayed binary feedback (good/bad) and their belief about their hit location. During this practice session, they received feedback about the correctness of their answer at the end of each trial. This practice session lasted 80 trials.

**Rule prediction main task.** The main task had the same structure as the practice task described above, except that participants did not receive feedback about the task state ('skill' or 'random') or the correctness of their choices. Participants completed 560 trials; they were allowed to take a short break every 80 trials. The duration of a given sequence of trials within one hidden task state, i.e., between two state change points, followed a Gamma distribution with parameters $k = 22.5$, $\theta = 1.33$, with a theoretical mean duration of 29.9 trials (true empirical mean across participants was $28.8 \pm 1.5$ trials). Based on performance, an extra reward was given to incentivise participants to hit the centre of the mole while correctly estimating the current task. Participants were told at

the beginning of the task that they would receive a reward point each time they correctly inferred the task state and obtained positive feedback. However, they were not informed about whether they were rewarded and how much until the end of the entire experiment. In the 'random' state, the reward rate was calculated as 50% of the correctly inferred trials (instead of considering actual empirical feedback). The real calculation was not revealed to participants to prevent probabilistic biases. The maximum extra reward was 3,000 JPY, and participants received an average of 2,122 ± 296 JPY.

**Model-free analyses**

**General behaviour.** Inference accuracy in the task was calculated as the ratio of correct answers ('skill', 'random') given the true hidden task state ('skill', 'random'). While the theoretical chance level was 0.5, we used a binomial test to determine at the individual level if a participant's data was overall above chance or not. Given 560 total trials, with a probability 0.5 of being correct on any given trial, the above-chance inference accuracy threshold was thus 0.54. As mentioned in the exclusion criteria, any participant whose accuracy was lower than this threshold was removed from further analyses. As representative examples, we selected two random participants and plotted the time series of their choices given the hidden rule. The time series of participants' choices was calculated as the moving average (backward window size = 15 trials) of the proportion of 'skill' choices. Overall, participants were similarly accurate in detecting skill and random states (see Table A in S1 Text).

**Contextual biases in choice behaviour.** To investigate whether participants displayed any choice bias based on the feedback received, we plotted the ratio of 'skill' choices, accuracy, and confidence separately for each feedback. Confidence ratings were z-scored within each participant. Wilcoxon signed-rank test with continuity correction was used to test the difference between the two feedback conditions. We used repeated measures two-way ANOVA to test the effect of feedback and task states and their interaction on inference accuracy or confidence. Wilcoxon signed-rank test with continuity correction was used to test the simple main effect of each factor when the interaction was significant. Because there were four pairwise comparisons (given two feedback and two task states), p-values were adjusted with false-discovery rate (FDR) correction.

**Task state change points and participants' behavioural adaptation.** To see how participants adapted their choice behaviour and confidence ratings to task state switches, we first aligned the data to the task change points. Next, we labelled the change points based on the direction of the switch, i.e., 'skill'→'random' and 'random'→'skill'. We extracted the four trials preceding the change point for each switch direction and the eight trials following the change point. We averaged trials at each time point across occurrences within participants. The twelve-trial time courses within participants were then used across participants to compute the population mean and standard error of the mean for each switch direction. To statistically analyse how the switch direction affected inference accuracy and confidence following a task state switch, we used repeated measures two-way ANOVA with factors time (trials from the switch) and switch direction, as well as their interaction. We further tested the difference in accuracy or confidence at each trial after the switch using a Wilcoxon signed-rank test with continuity correction, adjusted with FDR for multiple comparisons correction across the eight trials.

To investigate the underlying factors that caused participants to switch their choices, we plotted the subjective switching probability based on the previous choice ('skill', 'random') or confidence ('low', 'high') and the feedback ('positive', 'negative'). Note we binarised confidence into high and low levels through a median split within each participant. The effect on switch probability of the two factors, choice/confidence and feedback, and their interaction, was tested using repeated measures two-way ANOVA. We applied Wilcoxon signed-rank test with continuity correction and FDR adjustment to each pair if the interaction was significant.

**Hitting behaviour.** We investigated participants' mole-hitting patterns. This analysis examined whether participants might have changed where they hit the mole (consciously or unconsciously) depending on the feedback and inferred

task state. To this end, for each trial, we extracted the distance between the hit location and the centre of the mole. We normalised the obtained value by the within-participant median distance. Using repeated measures two-way ANOVA, we tested the difference in the distance between the current and next trial. If the interaction was significant, we applied a Wilcoxon signed-rank test with continuity correction and FDR adjustment for multiple comparisons to each pair.

## Model-based analyses

We developed a leaky evidence-accumulation model with separate error sensitivity and constant terms for each feedback type (positive vs. negative [56,57]). Here, we defined prediction errors following the intuition that participants have an internal estimate of how well they can hit the mole's centre, which depends on a subjective threshold parameter. This estimate helps them make an internal decision on whether they were more likely to have hit the centre or the edge of the mole.

**Full model.** The model features a leaky decision evidence accumulator, updated on each trial based on an error signal resulting from the mismatch between the binarised hit location (centre, edge) and the feedback (positive, negative). To judge if the hit location was in the centre, the model assumes participants have an internal, subjective estimate of the size of the central area, i.e., a threshold parameter that reflects the hypothetical separation between the central and the outer regions. Accordingly, there will be no error if the hit is considered in the centre and positive feedback is obtained, or if the hit is outside the central area and negative feedback is obtained. Other cases (such as hitting the centre and obtaining negative feedback) will result in an error signal. In this context, the model is closest to the ground truth when the threshold parameter is identical to the true threshold used in the task to determine feedback in the skill state. Thus, participants will be sub-optimal if they overestimate (subjective threshold > true threshold) or underestimate (subjective threshold < true threshold) how well they hit the centre. The model evaluates if the agent hits the centre as follows:

$$f_{centre} = 1, \ if \ d_t < \theta, \ otherwise \ 0 \tag{1}$$

where $t$ represents the trial, $d$ is the distance between the hit location and the centre of the mole, and $\theta$ is the free parameter representing the subjective threshold.

The error $e_t$ is calculated using the centre-edge estimate and the feedback $o_t$ obtained at trial $t$:

$$e_t = |f_{centre} - o_t| \tag{2}$$

The model accumulates the error through a hidden variable $z_t$, and outputs the choice probability $P$:

$$z_t = \alpha_{pos/neg} e_t + \gamma z_{t-1} \tag{3}$$

$$P(\hat{s}_t = random) = sigmoid(z_t + \beta_{pos/neg}) \tag{4}$$

where $\hat{s}_t$ is the choice, and ($\alpha_{pos}$, $\alpha_{neg}$, $\beta_{pos}$, $\beta_{neg}$, $\gamma$) are free parameters; $\alpha$ represents the error sensitivity for the current error observation (differently for positive and negative feedback), $\beta$ is a constant term modulating the decision boundary on the inference choice (differently for positive and negative feedback), and $\gamma$ is the retention factor. Thus, the full model has six free parameters in total. Note that, for simplicity, we did not explicitly include $d_t$ in the model's computation of choice probability, because $d_t$ would produce an additional non-linearity (the threshold being elsewhere than halfway between centre and edge) which would require additional parameters.

Finally, we calculate negative entropy as:

$$-H(P) = P(\hat{s}_t = random) \cdot log(P(\hat{s}_t = random)) + P(\hat{s}_t = skill) \cdot log(P(\hat{s}_t = skill)) \qquad (5)$$

We used negative entropy z-scored within participants as a model measure of confidence to compare with participants' actual confidence ratings. However, it is important to note here that confidence was not explicitly modelled (i.e., it was not part of the model fitting procedure) and remained a simple additional read-out.

**Parameter estimation.** We fit the model separately for each participant. The model takes the feedback sequence as input, and the trial-wise hit location is expressed as the distance from the centre, which outputs the choice probability. Consequently, we computed the cumulative negative log-likelihood over all time points within participants. For each participant, we estimated all free parameters ($\alpha_{pos}$, $\alpha_{neg}$, $\beta_{pos}$, $\beta_{neg}$, $\gamma$, and $\theta$) simultaneously by minimising the negative log-likelihood with a numerical minimisation method. The minimisation was done with the *constrOptim* function in R using the Nelder-Mead algorithm. The error sensitivity $\alpha$ and constants $\beta$ were constrained within the range [-20, 20], the retention factor $\gamma$ within the range [0, 1], and the threshold $\theta$ within the range [0, 43] (note that 43 represents the smallest integer superior to the farthest hit location across all participants). We fit the parameters with ten different sets of random initialisations and used the best result as the final initialisation to fine-tune the parameters and perform model comparisons.

**Alternative models and model comparisons.** We compared the full model with a range of alternative, simpler models. These were created by simplifying the main full model, such as removing specific parameters or the positive/negative asymmetry. The full list of models is displayed in Table B in S1 Text.

All models were fitted to each participant's data using the same procedure described above for the full model. The randomly initialised parameters were fitted using the Nelder-Mead algorithm with a constrained range and fine-tuned using the best ones as starting points. We calculated the Akaike Information Criterion (AIC) and Bayesian Information Criterion (BIC) for each participant and model and used them to evaluate goodness of fit. We used a Wilcoxon signed-rank test to test the AIC difference between each model and the full model (Fig E in S1 Text).

**Parameter recovery.** We performed a parameter recovery analysis to validate the model's reliability. First, we defined 100 sets of ground truth parameters; each set was randomly generated by sampling each parameter from a normal distribution whose mean and standard deviation were calculated from participants' estimated parameter values. Next, we generated a sequence of simulated choices for each parameter set, using the original sequence of hit locations and feedback as input to the model. As a result, we obtained 5100 simulated sequences (51 original sequences X 100 simulations each). The model fitting procedure was applied to each simulated sequence, thus obtaining pairs of ground truth and estimated parameters. Finally, we calculated Spearman's rank correlation coefficient for each parameter and generated the corresponding confusion matrix (Fig H in S1 Text). Results indicate good recovery performance, with correlations in the range of $r = 0.74$, 0.88. Instead, correlations between pairs of unrelated parameters resulted in values near zero, in the range $r = -0.06$, 0.09.

**Model simulations.** We performed additional model simulations with arbitrary parameter settings to validate the model findings on key behavioural indicators [58]. For all these simulations, the model sequentially computes $p_t$, the probability of choosing "random" for each trial based on a given parameters set, using the participants' feedback and hit locations (distance from the mole centre) as inputs. As a result, the model outputs a time series of probabilities for each participant. Additionally, trial-by-trial entropy was calculated using equation (5). To prevent issues with logarithm calculations when $p_t = 1$, $p_t$ was clipped to a maximum of $1 - 10^{-5}$. Accuracy for each trial was defined as $p_t$ when the true state was random and $1 - p_t$ when the true state was skill. The negative entropy values were normalised within participants and used as a proxy for confidence.

Our objectives were to examine the impact of the asymmetry in parameter estimates for error sensitivity and constant terms and of threshold overestimation on performance. We thus focused on two sets of parameters: the threshold ($\theta$) and the error sensitivity/constant term parameters ($\alpha$ and $\beta$).

To this end, we compared two conditions for the threshold modifications to assess overconfidence in hitting ability. The baseline condition used a θ value estimated through model fitting based on the observed behaviour, which we found to systematically overestimate the true threshold. In the true threshold condition, the θ parameter was set to each participant's true threshold, as determined by the task specifications, providing a benchmark that would reflect optimal or unbiased performance. To evaluate the effect of asymmetrical error sensitivity and constant terms, we introduced two additional conditions alongside the baseline for comparison, which used the fitted α and β from the model fitting. In the single parameter condition, we removed the asymmetry by averaging the parameter estimates; expressly, α was set to the mean of the positive and negative α values, and β was set to the mean of the corresponding β values.

Based on these conditions, we conducted eight simulations as follows: (i) with subjective threshold, feedback-dependent error sensitivity and constant term (the original full model); (ii) with true threshold, feedback-dependent error sensitivity and constant term; (iii) with subjective threshold, feedback-dependent error sensitivity and single constant term; (iv) with true threshold, feedback-dependent error sensitivity and single constant term; (v) with subjective threshold, single error sensitivity and feedback-dependent constant term; (vi) with true threshold, single error sensitivity and feedback-dependent constant term; (vii) with subjective threshold, single error sensitivity and constant term; (viii) with true threshold, single error sensitivity and constant term.

## Supporting information

**S1 Text. Supplementary Information file. Fig A.** Histogram of durations, measured in number of trials, of hidden task state sequences (mean = 28.8, standard deviation = 1.5). Each colour represents one participant (N = 51). **Fig B.** The task was calibrated such that participants would obtain roughly 50% positive scores in both skill and random states. Note that, nevertheless, the overall ratio of positive scores was significantly larger in the skill state compared to the random state at the group level (Wilcoxon signed-rank test, Z = 2.58, p = 0.01). **Fig C.** (1) Distribution of log-transformed reaction times in rule inference, divided by score (positive, negative). (2) Distribution of hit locations, expressed in pixels as the distance from the mole centre and divided by score (positive, negative). **Fig D.** Participants' hit location patterns are expressed as the distance of the hit from the mole centre. (1) distance from the centre divided by hidden task state (random, skill). (2) distance from the centre around task state switches, plotted from -4 trials before the switch to +8 trials after the switch. The trajectories are plotted separately for the two types of transition, from random to skill and skill to random. (3) cross-trial dynamics of participants' hit behaviour, separately for trials in which participants chose random or skill after obtaining a negative or positive score. The plot shows that participants adjusted their hit locations accordingly. After receiving a negative score, participants tended to hit closer to the centre (increased precision of hits). In contrast, after receiving a positive score, participants tended to hit further away from the centre (a relaxation of the precision). **Fig E.** Comparison of the full model against alternative models, in which specific parameters, or sets thereof, were fixed (e.g., in which the error sensitivity α was constrained to be score-independent). We used the Akaike Information Criterion (AIC) [59,60] to compare models. Note that lower AIC values mean better model fit. The plot shows each model's average AIC values (computed across participants). Wilcoxon signed-rank test, with FDR correction for multiple comparisons, demonstrated that the full model provided a significantly better fit for participants' data. It also showed that the subjective threshold parameter played a more important role than the constant term or the error sensitivity (both reflecting a positivity bias) due to the larger difference in AIC compared to the full model. The red arrow indicates the best-fit model, *** $P < 0.001$. **Fig F.** Quantitative correspondence between participants' behaviour and the model's simulated behaviour in terms of p(correct state inference). Each subplot represents one of the four conditions based on scores (negative, positive) and hidden task states ('random', 'skill'). From top left to bottom right: negative score, random task state (r = 0.96, $P < 10^{-10}$); positive score, random task state (r = 0.96, $P < 10^{-10}$); negative score, skill task state (r = 0.96, $P < 10^{-10}$); positive score, skill task state (r = 0.96, $P < 10^{-10}$). Each plot N = 51, circles represent individual participants, and solid/dotted lines represent the linear fit. **Fig G.** Inference accuracy, plotted by hidden task state and score. Simulations with specific parameter settings to evaluate

the effect of cognitive strategies on inference accuracy. We performed eight simulations based on three parameters, with two possible settings each. Across plots, the small circles represent participants' average inference accuracy, and the error bars are the standard error of the mean. The boxplots represent the simulation median and interquartile ranges. (A) Simulating the best fitting model with subjective threshold, asymmetric (score-dependent) error sensitivity, and asymmetric (score-dependent) constant term. This simulation is the same as reported in the main manuscript in Fig 2E. Note the close mapping between participants' behaviour and simulation results. (A, C, E, G) Simulations based on the model with the subjective threshold, modifying the error sensitivity and/or the constant term. (B, D, F, H) Simulations based on the model with the true threshold, modifying the error sensitivity and/or constant term. (H) Oracle agents use the true threshold and equal weights across positive and negative scores for error sensitivity and constant term. For all panels, simulation results are plotted as boxplots, with the box representing the median, first and third quartiles and whiskers representing the minimum and maximum of the data range, while participants results are plotted as circles representing the median and error bars representing the first and third quartiles. **Fig H.** Parameter recovery analysis, showing the confusion matrix with correlations between original (generated) parameters and recovered parameters. **Fig I.** Model recovery analysis. We first simulated data with each model using parameters within the range of fitted values from our participants' data, then fit each model to simulated data. All model comparisons displayed here were done using AIC. For all cases, the original ground truth model was consistently ranked as best fitting, in addition to the full model, since all alternative models are nested versions of the full model in which a parameter has been set to a fixed value. **Fig J.** Inference accuracy, plotted around hidden task state switches. Simulations with specific parameter settings to evaluate the effect of cognitive strategies on inference accuracy. We performed eight simulations based on three parameters, with two possible settings each. Across plots, the small circles represent participants' average inference accuracy, and the error bars represent the standard error of the mean. The boxplots represent the simulation median and interquartile ranges. (A) Simulating the best fitting model with subjective threshold, asymmetric (score-dependent) error sensitivity, and asymmetric (score-dependent) constant term. This simulation is the same as reported in the main manuscript in Fig 2E. Note the close mapping between participants' behaviour and simulation results. (A, C, E, G) Simulations based on the model with the subjective threshold, modifying the error sensitivity and/or the constant term. (B, D, F, H) Simulations based on the model with the true threshold, modifying the error sensitivity and/or constant term. (H) Oracle agents use the true threshold and equal weights across positive and negative scores for error sensitivity and constant term. **Fig K.** Correspondence between participants' confidence judgements and negative entropy of the model's decision output. Importantly, the model was not optimised based on confidence but solely based on the inference choices. The model's confidence was taken simply as the negative entropy of the decision output (since entropy signals the uncertainty in that decision). We evaluated the linear relationship with a linear mixed-effect model [Wilkinson formula *negative_entropy ~ confidence + (confidence | subjID)*]. The factor confidence was significant (estimate = 0.34, std = 0.026, $t_{50}$ = 12.86, $P < 0.001$). **Fig L.** Quantitative correspondence between participants' behaviour and the model's simulated behaviour in terms of p(correct state inference). Each subplot represents one of the four conditions based on scores (negative, positive) and hidden task states ('random', 'skill'). From top left to bottom right: negative score, random task state (r = 0.49, $P < 0.001$); positive score, random task state (r = 0.69, $P < 10^{-4}$); negative score, skill task state (r = 0.52, $P < 0.001$); positive score, skill task state (r = 0.26, $P = 0.069$). Each plot N = 51, circles represent individual participants, and solid/dotted lines represent the linear fit. **Fig M.** Confidence (negative entropy), plotted by hidden task state and score. Simulations with specific parameter settings to evaluate the effect of cognitive strategies on confidence. (A) Simulating oracle agents, using the true threshold and equal weights across positive and negative scores for error sensitivity and bias. Confidence shows minimal variation across hidden task states and scores. (B) Simulating agents with the true threshold but with a score-dependent error sensitivity and bias. Confidence again displays the original effect of scores, with higher confidence for positive scores than negative ones. (C) Simulating agents using the subjective threshold and score-independent error sensitivity and bias. Confidence shows the opposite pattern, being lower for positive scores and higher for negative scores and overall lower in skill compared to

random task state. (D) Simulating agents using the true threshold and inverted error sensitivity and biases (instead of the original weights being larger for positive than negative scores, weights are now larger for negative than positive scores). In this case, the confidence pattern reverses entirely compared to the original result, with higher confidence for negative than positive scores, regardless of task state. For all panels, simulation results are plotted as boxplots, with the box representing the median, first and third quartiles and whiskers representing the minimum and maximum of the data range, while participants results are plotted as circles representing the median and error bars representing the first and third quartiles. **Fig N.** Metacognition. (A) Formal meta-d' analysis for each task state (random and skill). Participants had equal metacognitive capacity across states. HMeta-d was used to calculate meta-d' [61]. (B) Formal meta-d' analysis based on the score obtained (negative, positive). Participants had equal metacognitive capacity across score types. (C) Area under the curve (AUC) analysis separately for each score-task state combination. Note that because we are now looking at each condition separately, it is no longer possible to compute meta-d' (which requires data from both responses within each condition of interest). We thus computed AUC on the confidence-choice response operating curve. Note the steep drop in AUC (metacognitive ability) specific to the negative scores in the skill task state. (D) Showing the underlying plot (C) data, notice the absence of a difference in confidence between correct and error trials, specifically after the negative scores in the skill task state. For all boxplots, the box represents the median, first and third quartiles, and whiskers represent the minimum and maximum of the data range; scatter plots represent participants' individual data points. **Fig O.** Time series of participants' confidence judgements (top) and model confidence (bottom) around hidden task state switches (e.g., random→skill or skill→random). The central line represents the mean across participants, and the error bars represent the standard error of the mean. Confidence was first z-scored and averaged within each participant/simulation. In both cases, the random condition leads to a higher average confidence than the skill condition. While this effect was stronger in the model results, it was also clearly present in participants' data. **Fig P.** Confidence (negative entropy), plotted around hidden task state switches. Simulations with specific parameter settings to evaluate the effect of cognitive strategies on inference accuracy. We performed eight simulations based on three parameters, with two possible settings each. Across plots, the small circles represent participants' average inference accuracy, and the error bars are the standard error of the mean. The boxplots represent the simulation median and interquartile ranges. (A) Simulating the best fitting model with subjective threshold, asymmetric (score-dependent) error sensitivity, and asymmetric (score-dependent) constant term. This simulation is the same as reported in the main manuscript in Fig 2E. Note the close mapping between participants' behaviour and simulation results. (A, C, E, G) Simulations based on the model with subjective threshold, modifying the error sensitivity and/or the constant term. (B, D, F, H) Simulations based on the model with the true threshold, modifying the error sensitivity and/or constant term. (H) Oracle agents use the true threshold and equal weights across positive and negative scores for error sensitivity and constant term. **Fig Q.** The plot shows the linear relationship between the perceptual distortion (represented as the ratio between the subjective and true thresholds) and the strength of the positivity/confirmation bias (defined as the difference between the error sensitivity for positive vs negative outcomes). We used robust regression to evaluate the strength of the relationship (slope = 2.84 ± 1.40, $t_{49}$ = 2.02, $P$ = 0.049). Dots represent individual participants, the line represents the linear fit, and the red asterisk indicates $P < 0.05$. **Fig R.** The plot shows the linear relationship between the individually calibrated true threshold computed as the median of all hit locations during the score prediction task, and the average score prediction error. A larger threshold means participants hit, on average, further from the centre of the mole, and a positive error means participants overestimated their ability to hit the centre. Thus, the x-axis represents motor ability, while the y-axis represents bias in the subjective evaluation of motor ability. We used robust regression to evaluate the strength of the relationship (slope = 2.18, $t_{49}$ = 4.58, p < 0.001). Dots represent individual participants, the line represents the linear fit, and the red asterisk indicates $P < 0.001$. **Table A.** Confusion matrix of task state and choices. Each cell is the ratio of choices within the relevant task state. The ratios within each task state sum to one within participants and are reported as mean ± STD computed across participants. Wilcoxon signed rank test on diagonal elements, i.e., true positive and true negative rates: Z = 1.87, P = 0.061. **Table B.** List of models used in the

computational analysis of behaviour. The full model, highlighted in bold, provided the best fit across participants (lowest AIC, BIC, LL). **Text A.** Description of task instructions provided to participants.
(PDF)

## Acknowledgments

We thank Peter Dayan for comments on an earlier draft, Nizar Mathli and Shihab Ahmed for help with task preparation, and Kaori Nakamura for help with participants' recruitment.

## Author contributions

**Conceptualization:** Michael Taylor, Benedetto De Martino, Aurelio Cortese.

**Data curation:** Naoyuki Okamoto.

**Formal analysis:** Naoyuki Okamoto, Takatomi Kubo, Shin Ishii, Benedetto De Martino.

**Funding acquisition:** Aurelio Cortese.

**Investigation:** Aurelio Cortese.

**Methodology:** Naoyuki Okamoto, Benedetto De Martino, Aurelio Cortese.

**Project administration:** Benedetto De Martino, Aurelio Cortese.

**Software:** Naoyuki Okamoto, Michael Taylor.

**Supervision:** Takatomi Kubo, Shin Ishii, Benedetto De Martino, Aurelio Cortese.

**Validation:** Aurelio Cortese.

**Visualization:** Naoyuki Okamoto, Benedetto De Martino, Aurelio Cortese.

**Writing – original draft:** Naoyuki Okamoto, Benedetto De Martino, Aurelio Cortese.

**Writing – review & editing:** Naoyuki Okamoto, Takatomi Kubo, Shin Ishii, Benedetto De Martino, Aurelio Cortese.

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
