## [Decision Letter · Decision Letter 0]

2 Jul 2025

Blaming Luck, Claiming Skill: Self-Attribution Bias in Error Assignment

PLOS Computational Biology

Dear Dr. Cortese,

Thank you for submitting your manuscript to PLOS Computational Biology. After careful consideration, we feel that it has merit but does not fully meet PLOS Computational Biology's publication criteria as it currently stands. Therefore, we invite you to submit a revised version of the manuscript that addresses the points raised during the review process.

Please submit your revised manuscript within 60 days Sep 01 2025 11:59PM. If you will need more time than this to complete your revisions, please reply to this message or contact the journal office at ploscompbiol@plos.org. Please include the following items when submitting your revised manuscript:

We look forward to receiving your revised manuscript.

Kind regards,

Zhaolei Zhang

Section Editor

PLOS Computational Biology

Zhaolei Zhang

Section Editor

PLOS Computational Biology

**Journal Requirements:**

At this stage, the following Authors/Authors require contributions: Naoyuki Okamoto, Michael Taylor, Takatomi Kubo, Shin Ishii, and Benedetto De Martino. Please ensure that the full contributions of each author are acknowledged in the "Add/Edit/Remove Authors" section of our submission form.

5) We have noticed that you have uploaded Supporting Information files, but you have not included a list of legends. Please add a full list of legends for your Supporting Information files after the references list.

2) If any authors received a salary from any of your funders, please state which authors and which funders..

7) Please ensure that all of the funders and grant numbers match between the Financial Disclosure field and the Funding Information tab in your submission form. Note that the funders must be provided in the same order in both places as well. Currently, the Financial Disclosure states there was no funding received.

**Reviewers' comments:**

Reviewer's Responses to Questions

**Comments to the Authors:**

Reviewer #1: The paper continues a recent trend in tackling a question that is important, but challenging, and about which more research is needed. I appreciate the choice of task and the inclusion of confidence estimates in the measures collected. Overall I have one major concern, which is that I feel the claim regarding a perceptual bias underlying self-serving attribution, which is repeatedly made, is not, in the current version of the manuscript, adequately supported. I believe more clarity is needed in the argumentation, especially in describing the modelling and how it supports the claim.

I provide more details about this below, together with some additional minor comments and suggestions.

Substance:

1. I think describing theta as modelling participants' perception of their own motor ability needs some discussion, since the threshold, as it is, can also be interpreted as capturing beliefs about how stringent or lax the task requirements are. Indeed this is the more natural interpretation, given the way the model is formulated, and especially given the fact that the model assumes that the distance between target center and hit location is known to participants (assuming I understood correctly). Teasing apart skill and task difficulty is of course very tricky, but there is a conceptual distinction between motor ability (being able to hit close to the center) and task requirements/ difficulty (what distance to center is good enough), as well as between perception of motor ability, i.e. distance from hit location to target center and inference about task requirements, i. e. the threshold for 'good' performance. I think these points needs more in depth discussion. Given this, I am not convinced that the modelling results really can be interpreted to support the claim, which is repeatedly made, that biases operate (also) at a "basic perceptual level".

2. The authors frame the modelling as designed to disentangle between two "fundamentally different underlying cognitive processes" - a perceptual one and a feedback processing one. But I think it is not sufficiently clear how exactly this question translates into their modelling approach. I would suggest to more clearly point this out when describing the parameters, and when reporting the model comparison results. As it is, model comparison is presented more as something performed for completeness than because it is essential for answering the question, which I think can be confusing.

5. The authors state that 'Thus the parameter fitting highlighted the overall tendency for positive scores to lead to skill choices', because alpha_pos was larger than alpha _neg and beta_pos was more negative than beta_neg. However, this doesn't seem to be quite the case, since the two parameters push the decision in different directions: while more negative beta means push towards skill, larger positive alpha (which is the case here) increases z and therefore pushes the decision towards random (again, assuming I understood correctly the model). And there seems to be a negative correlation between alpha and beta for each feedback type, according to figure 2D. I think it would be useful to discuss this in more depth.

I think this will also be relevant for the paragraph about the alpha parameter in the discussion, which I found a bit confusing. First of all, the authors state that Zamfir & Dayan's finding about participants updating their beliefs more after outcomes attributed internally than after those attributed externally aligns with the higher alpha for positive scores, which they say 'indicate that people tend to credit themselves for successes more readily than attribute failures to their actions'. If I understand the model correctly, this is now quite what the alpha parameter indicates: this is a parameter showing how much participants weigh evidence in favour of 'random', which would imply the opposite of what they claim in this paragraph.

It might also be relevant to mention here that Zamfir & Dayan found a larger learning rate for negative than for positive feedback, which is different from the authors' results.

Mid level:

1. The authors report, in the main text, that participants were above chance in detecting the condition (Skill vs random). However the interpretation of this information changes when one finds out that they excluded people who were below chance in detecting the condition (as per the methods). I think making this claim is to some extent misleading; at the very least I believe the authors should clearly report the exclusion criteria together with the statement about the inference accuracy in the main text.

2. The fact that inference accuracy was higher following positive scores is surprising, assuming the task was designed such that participants would get positive and negative feedback in roughly half the trials in skill, and with matching frequency in random. The authors address this by performing an ANOVA analysis, but I think it helps to have finer grained information about what feedback is given in what proportion of trials, in both conditions, in the main text. This information is indeed in the supplementary materials, but the way it is mentioned, very in passing, in the main text, seems a bit misleading to me: towards the end of the first paragraph of the introduction the authors say 'The task was calibrated such that participants would obtain positive scores with similar probabilities in both task states', which seems to imply this calibration was successful. However supplementary Fig 2 shows this was not actually the case: there was in fact a significant difference in the mean at the population level, and the distribution across subjects in the two conditions was qualitatively very different. Given the large amount of variability in the proportion of positive feedback in the skill condition, I think it would be useful to see if there is any relationship between the proportion of positive feedback in the skill condition and the proportion of correct inference across subjects. Because the pattern or feedback x state interaction for accuracy -higher accuracy after negative feedback in the random state and higher accuracy after positive feedback in the skill state- is compatible with a pattern where participants would roughly choose 'skill' after positive feedback and 'random' after negative feedback.

2. I think in the current version of the manuscript, there can be some confusion for the reader as to what self-attribution bias is: it seems to be defined as equivalent to self serving bias, i.e. the tendency to attribute mistakes to outside causes and successes to oneself. However in the next paragraph, the authors say it has been linked with depression, which is confusing: it would be better, I think, to make clear that depression has been linked with a reduced self serving bias. Then the authors quote Zamfir & Dayan as proposing self-attribution might give rise to learned helplessness; however this is not what the paper proposes. On the contrary, Zamfir & Dayan found a self-seving bias which might have a protective effect, counterbalancing higher learning from negative feedback.

2. The instructions and information given to participants in this context are likely to matter a lot: as the authors say, 'whack the mole' is generally perceived to be a skill task. Therefore it matters how participants are informed about the random condition. The methods are not clear and detailed enough to find out exactly how participants were instructed. Also: I think describing feedback as 'positive or negative score' might suggest numerical values, rather than 'Good' or 'Bad'. I think it would be better to use a different term, in order to avoid confusion.

3. I think the description of the model and assumptions it relies on (e.g that participants have access to the true distance between hit location and target center, if I understood correctly) needs to be clearer in the main text. As it is, I don't think one can interpret the results reported in the main text about the model parameters without reading the methods for additional details. I have two suggestions here: 1. to walk the reader through what the model does on every trial (ex. the participant keeps an estimate of x, transforms that into y, which, together with parameter k, controls action; once they receive feedback they compute z, and update x with z weighted by parameter b etc) And or 2. explicitly say what the different parameters do: e.g. a high value of parameter A means that subjects will do x more than if the value is low etc. This does not need to be very long, and would make the text more readable.

Further minor notes about the model description:

-it would be helpful to clearly state what the evidence being accumulated in the model is about: i.e. that it is treated in the model as evidence about the task being in the 'random' state.

-in the model graph in figure 2B, I find the labelling of variables as 'model' vs 'observed' somewhat confusing: s_t is gray but is not in the model variables rectangle, s^hat_t is in the model variables rectangle, but not in the observed variables, even though it is observed by the experimenter.

Low level:

1. Second phrase in the third paragraph of the introduction, starting with 'Similarly, in educational settings': the authors seem to want to switch from the students' perspective to teachers' perspective, but something is missing.

2.Supplementary figure 3B: does this only refer to the 'skill' condition? It should, I think, but it is not clearly stated.

3. Last paragraph before section "Behavioural signatures of self-attribution bias and computational modelling": 'The magnitude of the difference was larger after a positive score compared to a negative score, whichever rule they chose." This is not clear to me: is this the difference between the distance to center in trial t+1 vs trial t? Because it doesn't seem to match the pattern in Sup Fig 4C.

4. Third paragraph of the 'Behavioural signatures of self-attribution bias and computational modelling' section: 'we developed a computational model of our task in which scores derive from the joint effect of the hit location and the hidden task state'. I think it should be made clearer that this is a model of subjects' beliefs about how scores are derived.

Reviewer #2: See the attached document.

Reviewer #3: The paper touches on the question of attribution and its impact on further decision making. It uses a new task with skill, and random, hidden states, which differ in the way that feedback is given to subjects. The results are well developed and laid out. The approach is rigorous. I very much enjoyed reading this paper and do hope it will be published on this outlet.

This being said, there are some points that should be addressed to get this study in its best shape to be shared with the rest of the community. I'm going to lay these out in somewhat of an order.

1.a It seems that previous, very relevant work in the area is only very briefly touched upon and related to -- this goes for Mancinelli et al (2021); and Zamfir and Dayan (2024). The author introduce their new task by

"To resolve this question, we incorporated several new key features into our experimental approach (Mancinelli et al., 2021; Zamfir & Dayan, 2022). First, we diverged from previous studies by employing a sensorimotor task rather than the more commonly used multi-armed bandit tasks, in which factors beyond mere choice play no role."

The citation of these works is awkward here. Further nuance in this discussion is warranted -- esp. in relation to the actual novel questions that the new task is needed for. For instance, neither of those two works actually used bare multi-armed bandit tasks, but the casual reader might infer so from the way the paragraph is laid out.

1.b The same goes for the discussion of the results in relation to these previous two studies. For example:

"A recent study on self-attribution bias revealed dynamic, reciprocal relationships between attributions and self-beliefs, supporting the attribution-self-representation cycle theory (Zamfir & Dayan, 2022). They show that participants were more likely to update their beliefs about their abilities when they attribute outcomes to themselves rather than external factors. This aligns with our findings, which show a higher error sensitivity ( ) for positive scores compared to negative scores, indicating that people tend to credit themselves for successes more readily than attribute failures to their actions"

this felt unclear to me. Related:

"These findings illustrate how erroneously attributing successes to personal skill and failures to external factors can significantly affect subsequent behavioural strategies, often resulting in future suboptimal decisions (Garrett & Daw, 2020)."

This seems a point where to discuss Mancinelli et al. -- since they show that internality is linked to learning from failure.

2.a One of the main points of the paper is to show that altered performance perception lie at the origin of people's self-attribution biases. The authors could do a slightly better job at proving this. As far as I understand, the main point is in the paragraph:

"Comparisons between the full model and alternative, simpler versions (e.g., with a single score- independent error sensitivity or constant term, true threshold, and no retention factor) confirmed that the full model better accounted for participants’ choice strategies (see model comparison results in Supplementary Fig. 5). In addition, the model comparison further highlighted that the perceptual inflation (subjective threshold ) played a larger role than overall positivity bias (error sensitivity and constant term ) in determining behaviour, given the larger AIC difference from the full model when this parameter was fixed to the true threshold value (Supplementary Fig. 5)"

For at least the two models quoted in the last sentence, then, it would probably be good to perform model recovery in terms of the average AIC scores quoted. Either way, it would be good to point readers to exactly where it is shown that "perceptual inflation (subjective threshold ) played a larger role than overall positivity bias (error sensitivity and constant term ) in determining behaviour". The sentence does not reference figures or tables.

2.b Further, the authors should comment on why models have been fit separately per subject. This should be at least touched upon since the standard nowadays is to model subjects as random effects.

3. It seems that the wording is a tad unfortunate on the task state inference snippet (Figure 1, panel B). Good and skill are positively valued terms; Bad (I assume) would fit better with random. A more neutrally affective way of putting it forward

(e.g. giving the participant an altered distance from the center for example) could have been an alternative. Can the authors clarify this and its potential impact on results?

4. Figure 4 appears to be chopped on the left hand side. Y-label is missing and panel letter is unreadable.

**Have the authors made all data and (if applicable) computational code underlying the findings in their manuscript fully available?**

Reviewer #1: Yes

Reviewer #2: Yes

Reviewer #3: Yes

PLOS authors have the option to publish the peer review history of their article (what does this mean? ). If published, this will include your full peer review and any attached files.

**Do you want your identity to be public for this peer review?** For information about this choice, including consent withdrawal, please see our Privacy Policy .

Reviewer #1: No

Reviewer #2: No

Reviewer #3: No

**Figure resubmission:**
---

## [Decision Letter · Decision Letter 1]

3 Oct 2025

PCOMPBIOL-D-25-00774R1

Blaming Luck, Claiming Skill: Self-Attribution Bias in Error Assignment

PLOS Computational Biology

Dear Dr. Cortese,

Thank you for submitting your manuscript to PLOS Computational Biology. After careful consideration, we feel that it has merit but does not fully meet PLOS Computational Biology's publication criteria as it currently stands. Therefore, we invite you to submit a revised version of the manuscript that addresses the points raised during the review process.

Please submit your revised manuscript within 30 days Dec 03 2025 11:59PM. If you will need more time than this to complete your revisions, please reply to this message or contact the journal office at ploscompbiol@plos.org. Please include the following items when submitting your revised manuscript:

We look forward to receiving your revised manuscript.

Kind regards,

Zhaolei Zhang

Section Editor

PLOS Computational Biology

Zhaolei Zhang

Section Editor

PLOS Computational Biology

**Additional Editor Comments:**

Dear authors, while two reviewers are satisfied with the revision, reviewer 1 still raised some valid points which can further improve the manuscript and the presentation. Please address them accordingly in the next round of revision. I look forward to receiving the revised manuscript shortly.

**Reviewers' comments:**

Reviewer's Responses to Questions

**Comments to the Authors:**

Reviewer #1: I appreciate the added nuance in the interpretation of results and discussion and the increased clarity in the presentation of the modelling. I believe the paper is substantially better.

I do, however, have two remaining content-related comments.

-First of all the authors refer to 2 aspects as additional evidence in favour of a perceptual contribution to the bias and against the interpretation of the theta parameter as encoding beliefs about task difficulty. These two aspects are: the drop in confidence near the inferred subjective threshold and the fact that participants who overestimated their performance in the score prediction task also had larger objective thresholds. I do not find this argument convincing, as both of these observations are in fact consistent with a threshold encoding beliefs about task difficulty.

Regarding the first one: if a participant believes feedback is good/bad based on an threshold chosen by the experimenter, then it is not surprising that their confidence will be lower near the value of the threshold they believe is used than far away from it.

Regarding the second one: if a participant believes the threshold for obtaining positive feedback is very stringent it is not surprising that they will attempt to hit with higher precision than if they believe the threshold to be lax.

-Second, the way the authors describe and interpret Supplementary figure 18, which they describe in the discussion (lines 590-591) as "underscoring the positive correlation between motor ability and perception of motor ability" seems slightly inaccurate. A positive correlation between motor ability and perception of motor ability would mean that a lower threshold(i.e. higher accuracy in motor performance) was associated with a smaller absolute deviation between predicted and actual score (i.e higher accuracy in perception of ability). This is slightly different from what the plot shows. The plot does show that a higher objective threshold (therefore lower motor accuracy) is associated with a more positive score prediction error. However, lower objective thresholds seem (plotting the absolute prediction error on the y axis would indicate whether this is indeed the case or not) to correspond to more negative, rather than smaller absolute value prediction error. This is not necessarily consistent with the way the figure is described. It is also not clear why high motor accuracy would be associated with underestimation of score, under the perceptual effect that the authors are arguing for.

Lower level: the phrase starting with "the task was calibrated.." on lines 135-139 is difficult to parse and contains typos.

Reviewer #2: I thank the authors for their meticulous responses, which greatly clarified my earlier concerns. I had initially misinterpreted the type of self-attribution bias they aimed to capture, thinking they sought to document this bias even when participants had accurate perceptions of their abilities. This had made their design difficult for me to follow. With the revised version, however, the theoretical story is much clearer. I also appreciate the extensive robustness checks and model recovery analyses, which are highly reassuring to me. Overall, they addressed my main concerns and I recommend acceptance based on the new version.

The only remaining comment concerns Supplementary Figure 2. From my understanding, the authors conducted a two-sided Wilcoxon signed-rank test with the null hypothesis of no systematic difference between the two states, and reported a p-value of 0.01. By conventional interpretation, this would indicate a statistically significant difference, whereas the figure legend states that “the ratio of positive scores was relatively similar across states.” I may be missing some nuance here, but I would appreciate clarification of how these statements are reconciled.

Reviewer #3: I'm satisfied with the responses from the authors.

**Have the authors made all data and (if applicable) computational code underlying the findings in their manuscript fully available?**

Reviewer #1: Yes

Reviewer #2: Yes

Reviewer #3: None

PLOS authors have the option to publish the peer review history of their article (what does this mean? ). If published, this will include your full peer review and any attached files.

**Do you want your identity to be public for this peer review?** For information about this choice, including consent withdrawal, please see our Privacy Policy .

Reviewer #1: No

Reviewer #2: No

Reviewer #3: No

**Figure resubmission:**
---

## [Decision Letter · Decision Letter 2]

25 Nov 2025

Dear Dr Cortese,

We are pleased to inform you that your manuscript 'Blaming Luck, Claiming Skill: Self-Attribution Bias in Error Assignment' has been provisionally accepted for publication in PLOS Computational Biology.

Best regards,

Zhaolei Zhang

Section Editor

PLOS Computational Biology

Zhaolei Zhang

Section Editor

PLOS Computational Biology

Reviewer's Responses to Questions

**Comments to the Authors:**

Reviewer #1: I am satisfied with the added nuance and caution in the interpretation of the results.

**Have the authors made all data and (if applicable) computational code underlying the findings in their manuscript fully available?**

Reviewer #1: Yes

PLOS authors have the option to publish the peer review history of their article (what does this mean? ). If published, this will include your full peer review and any attached files.

**Do you want your identity to be public for this peer review?** For information about this choice, including consent withdrawal, please see our Privacy Policy .

Reviewer #1: No

---

## [Editor Report · Acceptance letter]

PCOMPBIOL-D-25-00774R2

Blaming Luck, Claiming Skill: Self-Attribution Bias in Error Assignment

Dear Dr Cortese,

I am pleased to inform you that your manuscript has been formally accepted for publication in PLOS Computational Biology. Your manuscript is now with our production department and you will be notified of the publication date in due course.

With kind regards,

Livia Horvath
